# How precipitation intermittency sets an optimal sampling distance for temperature reconstructions from Antarctic ice cores

Thomas Münch[1], Martin Werner[2], and Thomas Laepple[1,3]

[1]Alfred-Wegener-Institut Helmholtz-Zentrum für Polar- und Meeresforschung, Research Unit Potsdam, Telegrafenberg A45, 14473 Potsdam, Germany
[2]Alfred-Wegener-Institut Helmholtz-Zentrum für Polar- und Meeresforschung, Bussestraße 24, 27570 Bremerhaven, Germany
[3]University of Bremen, MARUM – Center for Marine Environmental Sciences and Faculty of Geosciences, 28334 Bremen, Germany

**Correspondence:** Thomas Münch (thomas.muench@awi.de)

**Abstract.** Many palaeoclimate proxies share one challenging property: they are not only driven by the climatic variable of interest, e.g., temperature, but they are also influenced by secondary effects which cause, among other things, increased variability, frequently termed *noise*. Noise in individual proxy records can be reduced by averaging the records, but the effectiveness of this approach depends on the correlation of the noise between the records and therefore on the spatial scales of the noise-generating processes. Here, we review and apply this concept in the context of Antarctic ice-core isotope records to determine which core locations are best suited to reconstruct local-to-regional-scale temperatures. Using data from a past-millennium climate model simulation equipped with stable isotope diagnostics we intriguingly find that even for a local temperature reconstruction the optimal sampling strategy is to combine a local ice core with a more distant core $\sim 500$–$1000$ km away. A similarly large distance between cores is also optimal for reconstructions that average more than two isotope records. We show that these findings result from the interplay of the two spatial scales of the correlation structures associated with the temperature field and with the noise generated by precipitation intermittency. Our study helps to maximise the usability of existing Antarctic ice cores and to optimally plan future drilling campaigns. It also broadens our knowledge on the processes that shape the isotopic record and their typical correlation scales. Finally, many palaeoclimate reconstruction efforts face the similar challenge of spatially correlated noise, and our presented method could directly assist further studies in also determining optimal sampling strategies for these problems.

## 1 Introduction

The oxygen and hydrogen isotopic composition of firn and ice recovered from polar ice cores is a key proxy for past near-surface atmospheric temperature changes (Dansgaard, 1964; Lorius et al., 1969; Masson-Delmotte et al., 2008; Sjolte et al., 2011). Although the physical mechanisms that link local changes in temperature to the isotopic composition of precipitated snow are generally well understood (Dansgaard, 1964; Craig and Gordon, 1965; Jouzel and Merlivat, 1984) and can be modelled with general circulation models (Joussaume et al., 1984; Werner et al., 2011, 2016; Sjolte et al., 2011; Goursaud et al.,

2018), the quantitative interpretation of ice-core isotope variability, in terms of temperature variability, is complicated by second-order processes that influence the isotopic record, adding noise (Münch and Laepple, 2018).

Specifically, the isotopic record that is derived from an ice core is the result of a chain of processes: (1) atmospheric temperature changes along with (2) isotopic fractionation during the pathway from atmospheric moisture to precipitation, (3) the effect of variable and intermittent precipitation and finally (4) local depositional and post-depositional effects. As we outline in the following, each element of this chain can be associated with a typical spatial length scale over which it is correlated.

Atmospheric temperature variations drive the isotopic composition fractionation of the atmospheric moisture along its pathway to the final stage of precipitation (Dansgaard, 1964; Jouzel and Merlivat, 1984). The spatial coherence of the temperature-related isotopic signal in precipitation is hence determined by the spatial coherence of the variations of the atmospheric temperature field itself. Typical spatial decorrelation scales for temperature anomalies are on the order of $\gtrsim 1000\,\mathrm{km}$ (Jones et al., 1997), which implies that ice cores distributed on spatial scales below $\sim 1000\,\mathrm{km}$ should typically record a similar, i.e., correlated, temperature signal. However, the temporal variability of the isotopic composition in the local atmospheric moisture also depends on the variability of the atmospheric circulation, since different air masses may exhibit different source regions and distillation pathways (Schlosser et al., 2004; Sodemann et al., 2008; Birks and Edwards, 2009; Küttel et al., 2012). In addition, the isotopic composition profile across a deposited layer of snow will not directly reflect the temporal variability of the atmospheric isotopic signal due to the intermittent nature of precipitation (Schleiss and Smith, 2015). By this, the initial isotope signal is weighted with the amount of precipitation, which introduces bias (Steig et al., 1994; Laepple et al., 2011) and adds additional variability to the isotopic record (Persson et al., 2011; Casado et al., 2020). The latter two processes are linked to atmospheric dynamics and their typical spatial scales range from the mesoscale (i.e., tens of kilometres), driven by topography and orographic effects, to synoptic scales of hundreds of kilometres, associated with cyclonic activity and the movement of high and low pressure systems. Finally, in polar conditions, the precipitated snow does not directly settle but is constantly eroded, blown away, and redeposited. These depositional processes have been shown to give rise to stratigraphic noise in the isotopic record (Fisher et al., 1985; Münch et al., 2016; Laepple et al., 2016), which exhibits a small-scale decorrelation scale of a few metres (Münch et al., 2016). We note further that the final isotopic record is also influenced by potential exchange processes at the surface and by densification and diffusion within the snow and ice, which are, however, not within the scope of this article.

Both the effect of precipitation intermittency and stratigraphic noise constitute a significant relative contribution to the overall isotopic variability in form of noise: Around a deep drilling site in Dronning Maud Land, East Antarctica, stratigraphic noise was shown to amount to approximately 50 % of the total variance at the seasonal time scale (Münch et al., 2016), but quantitative estimates for other Antarctic regions are still missing. A similarly high relative contribution is expected from precipitation intermittency (Laepple et al., 2018), which probably has a larger impact further inland than compared to coastal regions (Casado et al., 2020; Hatvani and Kern, 2017).

The hierarchy of the different spatial scales of the processes influencing an isotope record determines the effectiveness of reducing the overall noise, since a reduction in the noise level by averaging records will depend on the spatial correlation scale of the different noise sources. For example, if an isotope record were only shaped by temperature variations and stratigraphic

noise, it would be sufficient to average records spaced only tens of metres apart, as this would ensure highly correlated temperature signals but uncorrelated stratigraphic noise between the records. However, comparing the correlation-based signal-to-noise ratios derived from nearby isotope records (Münch et al., 2016, 2017) with the signal-to-noise ratios estimated from analysing the records' temporal variability (Laepple et al., 2018) shows that the reproducibility on a local scale does not necessarily imply a climatic, i.e., temperature-driven, origin. Instead, the additional noise sources from circulation variability and precipitation intermittency are likely to exhibit larger decorrelation lengths than the stratigraphic noise (Laepple et al., 2018; Münch and Laepple, 2018). Taking this into account, we expect there to be an optimal length scale which lies in between the decorrelation scales of the local noise and of the temperature, and which results in a trade-off between averaging out atmospheric circulation and precipitation intermittency effects, while also ensuring a sufficient coherence in the recorded temperature signal.

The aim of the present study is to use data from a climate model equipped with stable isotope diagnostics to systematically study the different typical process scales – including those from atmospheric temperature variations, circulation variability, precipitation intermittency and the isotope–temperature relationship –, to determine the optimal spatial arrangement of ice-core locations which maximises the correlation with temperature at a specific target site. To address this problem we focus on target sites on the East Antarctic Plateau. Our results show that the average of multiple ice-core isotope records yields a higher degree of correlation with temperature when the sampled locations are spread across distances of $1000\,\mathrm{km}$ or more from the target site, than when they are all located close ($< 250\,\mathrm{km}$) to the target site. While these results may seem counterintuitive at first, we qualitatively explain their general features with a simple analytical model that uses the typical spatial correlation structures associated with the temperature and isotope fields, and with the noise generated by precipitation intermittency.

## 2  Data and methods

### 2.1  Climate model data

We use data from the past-millennium simulation (800–1999 CE; Sjolte et al., 2018) of the fully coupled ECHAM5/MPI-OM-wiso atmosphere–ocean general circulation model equipped with stable isotope diagnostics (Werner et al., 2016). This simulation is forced by greenhouse gases, volcanic aerosols, total solar irradiance, land use changes, and changes in the Earth's orbital parameters. The model's atmospheric component ECHAM5-wiso is run with a T31 spectral resolution ($3.75° \times 3.75°$) and with 19 vertical levels (Sjolte et al., 2018). Compared to observations, the climatological relationship between temperature and the precipitation isotopic composition is reproduced well by the model, but it is biased towards warm temperatures in the T31 setup and its isotopic composition is not depleted enough over Antarctica (Werner et al., 2011). These issues can be improved upon by using a higher spatial resolution (Werner et al., 2011); however, such a higher-resolution model is not needed for our study, since we are mainly interested in the relative variability between sites and not in the absolute temperature or isotope values. The full atmosphere–ocean model was compared to observational data and palaeoclimate records for two equilibrium simulations under pre-industrial and Last Glacial Maximum conditions (Werner et al., 2016), and the past-millennium simulation was used to reconstruct North Atlantic atmospheric circulation in combination with ice-core isotope data (Sjolte et al., 2018).

In this study, we use the 1200-year ECHAM5/MPI-OM-wiso time series of two-metre surface air temperature ($T_{2\mathrm{m}}$), precipitation ($p$), and oxygen isotopic composition in precipitation (the relative abundance of oxygen-18 to oxygen-16 istopes, denoted as $\delta^{18}$O) extracted from the total number of $442$ model grid cells that are available for the Antarctic continent (Münch and Werner, 2020).

## 2.2  Data processing

The model simulation output has a monthly temporal resolution, while ice-core isotope records typically exhibit an annual (or even lower) resolution. The latter is commonly achieved by averaging the isotopic data across annual layers of snow and ice, which are determined through a dating approach. The resulting annual isotopic composition data therefore include a weighting effect due to the intra-annual variability in the amount of precipitation. To account for this, we produce two versions of annual data from the monthly model output (Münch and Werner, 2020): (1) the two-metre temperature and oxygen isotopic
composition data are averaged to an annual resolution without any weighting (denoted as $T_{2\mathrm{m}}$ and $\delta^{18}$O in the following), and (2) the respective monthly data are averaged to an annual resolution including the weighting by the monthly precipitation amount (denoted as precipitation-weighted data $T_{2\mathrm{m}}^{(\mathrm{pw})}$ and $\delta^{18}$O$^{(\mathrm{pw})}$).

In extremely dry areas with very little precipitation amounts or high evaporation, numerical instabilities can occur for the modelled isotopic composition in precipitation, resulting in anomalously strong positive or negative spikes in the isotope time
series, which is also observed for the Antarctic data in our model simulation. We set a threshold of $4$ times the interquartile range of a time series above or below which data points are regarded as outliers, and apply it to every grid cell in order to filter outliers in the $\delta^{18}$O and $\delta^{18}$O$^{(\mathrm{pw})}$ time series. This approach removes $443$ anomalous annual values ($< 0.1\,\%$), out of which $435$ anomalies occur for the model year $970\,$CE.

## 2.3  Data analyses

### 2.3.1  General approach

The overarching aim of this study is to determine a set of locations from which the averaged model data optimally reconstruct the $T_{2\mathrm{m}}$ temperature time series at a *target site*, i.e., a specified model grid cell of interest. The optimal reconstruction is assessed by maximising the Pearson correlation coefficient ($r$) to the target site temperature. To define a spatial set, we combine a given number, $N_\ell$, of model grid cells, and vary $N_\ell$ and the distances of these locations relative to the target site. To derive
implications for actual ice-core studies, we use the $\delta^{18}$O$^{(\mathrm{pw})}$ time series at the locations as a surrogate for ice-core isotope records. We thus neglect stratigraphic noise and any further depositional or post-depositional effects on the isotopic record, and therefore our results represent an upper limit of the extent to which ice cores can reconstruct the climatic temperature signal in the atmosphere. In order to learn about how the different underlying processes affect the results and to isolate their contributions, we compare the results obtained for $\delta^{18}$O$^{(\mathrm{pw})}$ with those obtained for $T_{2\mathrm{m}}$, $T_{2\mathrm{m}}^{(\mathrm{pw})}$ and $\delta^{18}$O. In addition to
using only a single target site, we analyse several adjacent target sites in a given region to derive results that are relevant

on local-to-regional spatial scales. In the next section, we present the two main methods that we use to assess the optimal reconstructions.

### 2.3.2 Assessing optimal reconstructions

*Selecting optimal sites*

In a first approach, we select an optimal set of ice-core locations to reconstruct a target site's $T_{2\mathrm{m}}$ time series by sampling without replacement $N_\ell$ grid cells that lie within a selection circle of $2000\,\mathrm{km}$ radius around the target site, and then correlating the average $\delta^{18}\mathrm{O}^{(\mathrm{pw})}$ time series from these $N_\ell$ grid cells with the target site temperature. We perform this for different $N_\ell$ and determine the optimal set of cores for each value of $N_\ell$ from the maximum correlation value across all selection trials. For this, we either sample all possible combinations of grid cell locations within the selection circle, if the number of possibilities does

not exceed $10^7$, which effectively applies to all $N_\ell \leq 3$, or we randomly sample $10^7$ times from all the possible combinations.

*Optimal sampling structure*

In order to learn about the typical spatial scales associated with the processes that contribute to the overall temperature–isotope relationship, we aim to investigate how the reconstruction quality depends on the radial distances between the target site and the

locations of the ice-core network only, neglecting their angular positions. To do so, one could use the above random selection trials and bin them according to the distances of the selected locations relative to the target site. However, for $N_\ell > 1$ the number of possible grid-cell combinations quickly becomes much larger than the actual number of grid cells. In combination with the limited computation time, such an approach would likely result in uneven sample sizes for the available distance combinations for larger $N_\ell$, especially for distances farther away from the target site due to the radially increasing number of

grid cells.

Here, we instead use a second more general approach that ensures constant sampling of the entire available space of radial distance combinations, and which also reduces local effects in the climate model data and provides more stable correlation results. For a given target site, we define as sampling regions nine concentric rings around the target site with increasing radius in steps of $250\,\mathrm{km}$ (Fig. 1) and identify all grid cells that lie within each of these rings. The sampling of $N_\ell$ grid cells is

then implemented in the following two-step process: First, we determine all possible combinations of selecting $N_\ell$ rings with replacement. For every ring combination, we then apply the following second step: we sample one individual grid cell from each of the $N_\ell$ rings (see the examples in Fig. 1b for an illustration), extract from this grid-cell set the time series for a studied model variable, average them, and compute the degree of correlation of this average record with the target site temperature. This second step is iterated over the available number of grid-cell sets and we report the mean correlation across all analysed

grid-cell sets. For the iteration, we identify all possible grid-cell sets until $N_\ell = 2$; for $N_\ell \geq 3$, we resort to Monte Carlo sampling of the grid-cell sets due to computational reasons, for which we estimated a number of $10^5$ iterations to provide sufficient convergence of the results.

This approach provides insight into the average spatial structure of the correlation with the target site temperature for sampling $N_\ell$ locations from the model field depending on the radial distances of the locations, as given by the respective ring

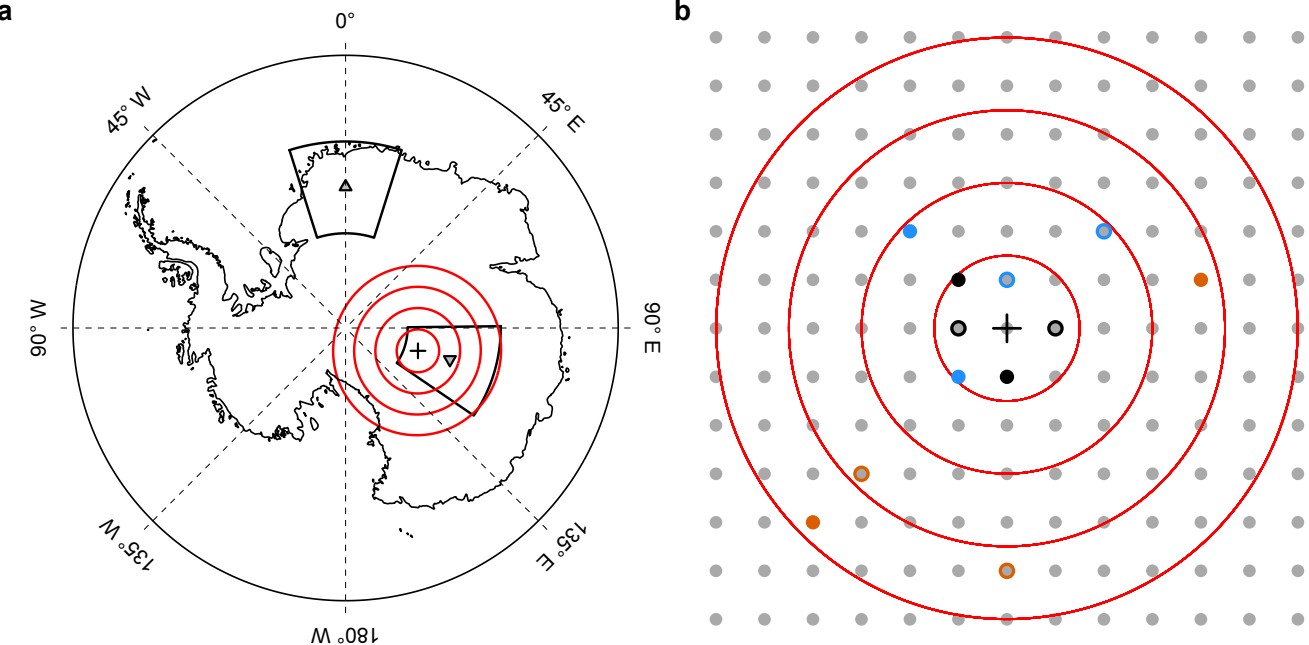

**Figure 1.** Conceptual sketch of the ring sampling approach. Around a given Antarctic target site (black crosses in **a** and **b**) we define consecutive rings of 250 km radial width (red lines in **a** and **b**). From the array of available model grid cells (grey points in **b**), we choose sets of grid cells which consist of $N_\ell$ cells and which are drawn from $N_\ell$ radial bins determined by a selected combination of rings. As an example for $N_\ell = 2$, possible grid-cell sets are shown for the cases of (i) combining the innermost ring with itself (grid cells marked black), (ii) combining the innermost ring with the second ring (grid cells marked blue), and (iii) combining the third and the fourth ring (grid cells marked orange). Also shown in (**a**) are our main study regions (black polygons) around the EDML (upward pointing triangle) and Vostok (downward pointing triangle) ice-core sites. The ring width of 250 km is chosen as a trade-off between high spatial resolution and the requirement that a sufficient number of grid cells lie inside each ring. Note that for aesthetic reasons, only four rings are displayed instead of the actually used nine rings and that the model grid is shown simplified as a regular grid in space.

midpoint radii. We denote this quantity as the *sampling correlation structure*. Note that in the one-dimensional case ($N_\ell = 1$), the sampling correlation structure is identical to what is often called the spatial correlation structure, i.e., the average correlation as a function of radial distance.

### 2.3.3   Study regions

To derive sampling correlation structures which are representative on a regional scale, we conduct the above analysis for
specific regions by successively using each model grid cell in the region as a target site and then averaging the resulting sampling correlation structures across these target sites.

We make use of this approach for two subregions of the East Antarctic Plateau, the Dronning Maud Land (DML) region in the Atlantic sector of the plateau and the Vostok region in the Indian Ocean sector, both of which include existing deep ice-core drilling sites as well as large arrays of shallower ice and firn cores. We define the DML region as the area of $\pm 17.5°$ longitude and $\pm 5°$ latitude around the European Project for Ice Coring in Antarctica (EPICA) DML site (EDML; 75° S, 0° E; Fig. 1a), consisting of 26 model grid cells. This region encompasses the site of the deep EDML ice core (EPICA community members, 2006; Alfred-Wegener-Institut Helmholtz-Zentrum für Polar- und Meeresforschung, 2016) and $> 50$ firn and shallow ice cores (Altnau et al., 2015). For the Vostok region, we choose an identical latitudinal and longitudinal coverage with respect to the Vostok station (78.47° S, 106.83° E; Fig. 1a), covering 30 model grid cells, and encompassing the sites of the deep Vostok and Dome C ice cores, of several shallower cores (Stenni et al., 2017), and of the new deep drilling project ("Little Dome C") where an ice core extending back more than one million years is envisaged (Passalacqua et al., 2018).

## 3 Results

### 3.1 Spatial scale of the temperature anomalies and the local temperature–isotope relationship

First, we asses the extent to which a single ice-core record, i.e., the annual isotope time series of an individual grid cell in the model simulation, is representative of the local and regional scale variability of the near-surface atmospheric temperature.

The temperature field over Antarctica in the climate model exhibits large scale coherent variations (Fig. 2a) with a clear two-part structure, which is roughly divided by the Transantarctic Mountain range: For most parts of the East Antarctic Plateau, the temperature field shows typical decorrelation lengths between $\sim 1500$ and $2500\,\mathrm{km}$, while the decorrelation lengths are notably lower with values ranging from $\sim 500$ to $1500\,\mathrm{km}$ for larger parts of the West Antarctic Ice Sheet and for the Antarctic Peninsula. Still, for perfect ice cores, i.e., assuming an ideal temperature proxy record that is only governed by local temperature variations, a single ice core would capture the temperature variability in both East and West Antarctic regions across hundreds of kilometres.

However, as simulated by the isotope-enabled climate model, actual single Antarctic ice-core isotope records only explain a low portion of the variations in the local temperature fields: Correlating the annual precipitation-weighted field of modelled $\delta^{18}\mathrm{O}^{(\mathrm{pw})}$ with the annual $T_{2\mathrm{m}}$ time series at the same grid cell results in generally low correlations (mean of 0.38), which across all analysed grid cells range from $\sim 0.1$ up to $\sim 0.57$ with $\sim 60\,\%$ of the correlations $\leq 0.4$ (Fig. 2b). The correlations are overall improved when the $T_{2\mathrm{m}}^{(\mathrm{pw})}$ time series is used instead of the $T_{2\mathrm{m}}$ time series (mean correlation of 0.53, range $\sim 0.1$ to 0.77; Fig. 2c), but with unaffected correlation values mostly in the coastal regions (Fig. 2d). This shows that precipitation intermittency is a major limiting factor for the local temperature–isotope correlation on the continental plateau, but is less important near the coasts due to higher and more regular snowfall amounts there (Casado et al., 2020).

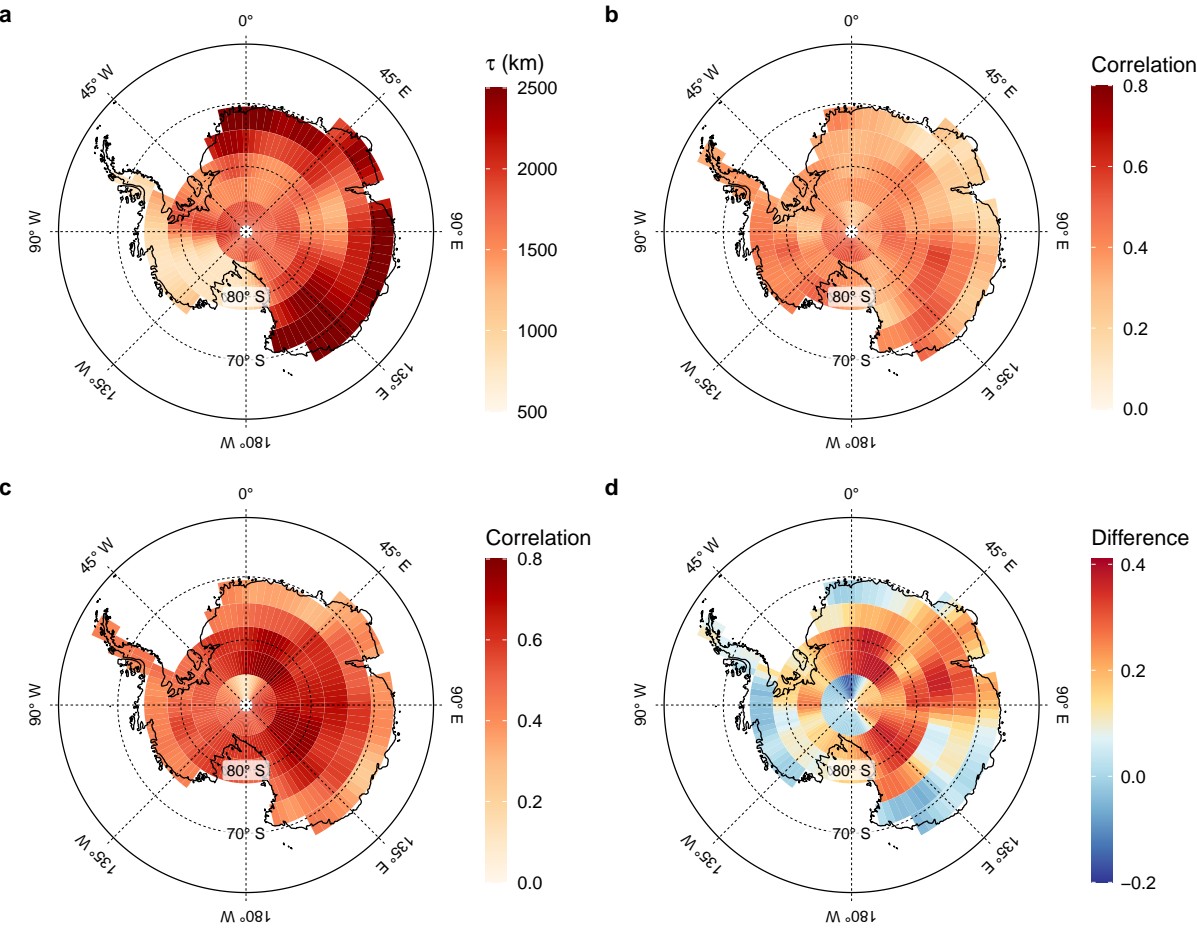

**Figure 2.** Temperature decorrelation lengths and local temperature–isotope relationship across Antarctica. (**a**) The temperature decorrelation lengths ($\tau$, in km) for each Antarctic model grid cell, estimated by fitting an exponential model to the correlation–distance relationship (cf. Eq. A4) obtained from correlating the local annual near-surface $T_{2m}$ time series with the respective temperature time series from all other grid cells. Note that only the continental grid cells were used for the fit. Although the decorrelation lengths show a strong partition between East and West Antarctica, they are larger than $1000\,\mathrm{km}$ at most locations. (**b**, **c**) The local correlations at each model grid cell between the annual time series of precipitation-weighted oxygen isotope composition and of (**b**) near-surface temperature and (**c**) precipitation-weighted near-surface temperature. The difference between the maps (**d**) clearly demonstrates that precipitation intermittency is a major limiting factor for the temperature–isotope relationship.

## 3.2 Spatial correlation to local temperature

In the next step, we investigate how a local temperature record correlates in space with the temperature itself and with the oxygen isotope composition. For this, we choose the EDML and Vostok drilling sites as target sites, and correlate the annual

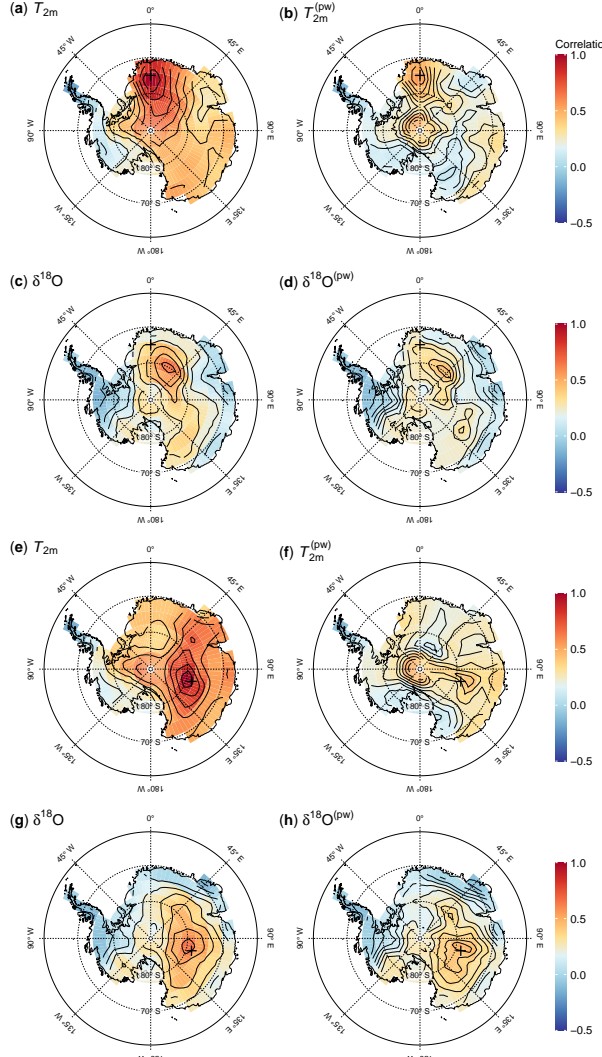

**Figure 3.** Spatial correlation to the temperature at the EDML and Vostok target sites. Shown are the correlations of the $T_{2m}$ time series at the target sites EDML (**a–d**) and Vostok (**e–h**) with, respectively, the spatial fields of temperature (**a, d**), precipitation-weighted temperature (**b, f**), oxygen isotope composition (**c, g**), and precipitation-weighted oxygen isotope composition (**d, h**). The target sites are marked with a black cross, black lines indicate correlation contour lines incremented in steps of $0.1$.

$T_{2m}$ time series at these target sites with the spatial fields of annual temperature and of annual $\delta^{18}O$, both unweighted and
195   weighted by the precipitation amount (Fig. 3).

We find that the correlation patterns with the temperature field itself are largely radially symmetric with respect to the target sites and decay uniformly with distance within the first couple of hundred kilometres from the target (Fig. 3a, e). However, for $\delta^{18}O$, and also partly through the effect of the precipitation weighting, radial asymmetry in the correlation patterns occurs. This

is in particular striking for the EDML target site. Here, the maximum in correlation with the $\delta^{18}O$ field is not centred on the target site but displaced by $\sim 1200\,\text{km}$ towards the southeast (Fig. 3c, d). Some spatial displacement in maximum correlation is also visible for the Vostok target site and the $T_{2m}^{(pw)}$, $\delta^{18}O$ and $\delta^{18}O^{(pw)}$ fields (Fig. 3f–h), but in different directions between $T_{2m}^{(pw)}$ and the oxygen isotope fields and much smaller than in the case of EDML. We also note that the correlation patterns for the $T_{2m}^{(pw)}$, $\delta^{18}O$ and $\delta^{18}O^{(pw)}$ fields still contain radially symmetric contributions with respect to the target sites, which are more pronounced for the Vostok than for the EDML target site.

### 3.3 Selecting optimal ice-core sites for temperature reconstructions

The above analyses have shown firstly that isotope records from single ice cores likely only capture a small portion of the local interannual temperature variability, suggesting that additional processes, such as precipitation intermittency, influence the isotopic signal and decrease the degree of correlation with the local temperature record. Interpreting these additional processes as noise raises the question of whether the correlation with temperature can be improved upon by averaging isotope records across space. In addition we have seen that the correlation of an oxygen isotope composition record with a local temperature record is not necessarily maximal at the location of the temperature recording, posing the question of how locations of isotope records should be spatially arranged with respect to the location of the temperature record in order to get the best correlation. To address these questions, we assume an ideal world in which the climate model data are a perfect surrogate for the true climate and proxy variations at each site, and set up the simple experiment of selecting and averaging $\delta^{18}O^{(pw)}$ records from grid cells within a $2000\,\text{km}$ circle around a target site (see Sect. 2.3.2 for details) to determine what spatial array of $N_\ell$ ice cores optimises the temperature correlation with the target site.

For our specific model simulation and specifying the EDML drilling site as the target site, we already know from Fig. 3a that the optimal location for a single ice core is not the local, i.e., target site grid cell, but should be a $\sim 1200\,\text{km}$ southeastward site. Choosing this more distant site increases the correlation with the target temperature from an $r$ value of $0.30$ for the local EDML site to a value of $0.44$ (Fig. 4a). Even more intriguingly, when we analyse the maximum correlations with the EDML target temperature for an average of three or five cores chosen from the $2000\,\text{km}$ selection circle (Fig. 4b–c), we find optimal locations that in both cases are scattered at significant distances around the target and which yield an even further increase in correlation ($r = 0.50$ for $N_\ell = 3$, $r = 0.52$ for $N_\ell = 5$). We obtain comparable results when the Vostok drilling site is specified as the target (Fig. 4d–f). The optimal single core would be at a location $\sim 190\,\text{km}$ west of Vostok ($r = 0.49$, compared to the local correlation of $r = 0.46$), and the optimal locations for averaging three or five cores all lie again scattered around the target without including it and result in a significant further increase in correlation ($r = 0.60$ for $N_\ell = 3$, $r = 0.63$ for $N_\ell = 5$).

We generalise these findings by considering each Antarctic model grid cell as a target site and determining in each case the ice core location that results in an optimal correlation with the target site. As in the above EDML case study, about half of the optimal locations for a single ice core are situated at distances between $500$ and $1500\,\text{km}$ from the respective target sites, while only about $10\,\%$ lie within $500\,\text{km}$ from the targets. We note that this distribution might be affected by the number of available sampling points (i.e., model grid cells) per distance bin which increase with increasing distance from the target site.

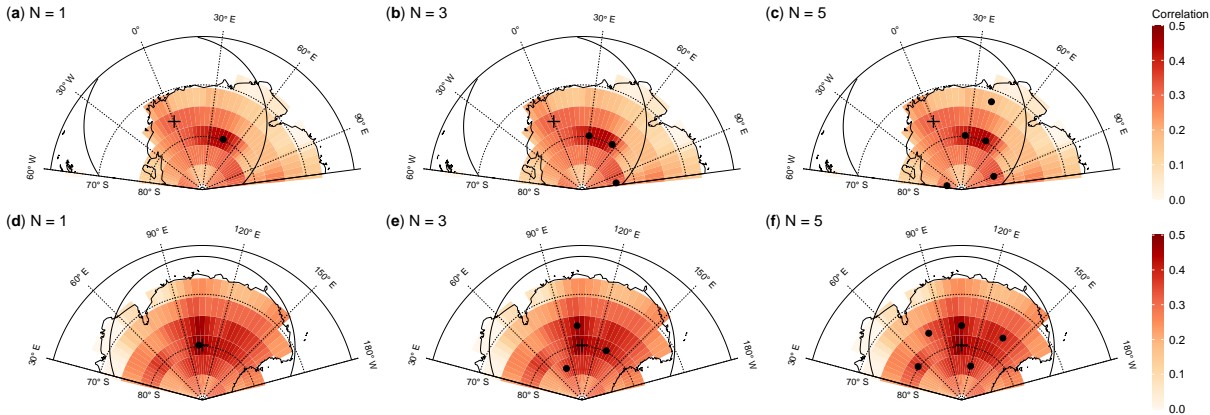

**Figure 4.** Selecting ice-core locations that optimally reconstruct interannual temperatures at the EDML and Vostok drilling sites. The maps show the correlation coefficient in the climate model data between the annual temperature time series at the target sites (black crosses) EDML (**a**–**c**) and Vostok (**b**–**f**) with the time series fields of precipitation-weighted oxygen isotope composition ($\delta^{18}O^{(pw)}$). Filled black points denote those grid cells that yield the maximum correlation between the target site temperature and the $\delta^{18}O^{(pw)}$ time series from either selecting a single grid cell ($N_\ell = 1$; **a, d**) or from averaging across $N_\ell = 3$ (**b, e**) or $N_\ell = 5$ (**c, f**) grid cells, obtained from iteratively selecting sets of $N_\ell$ grid cells from within a selection circle of 2000 km radius around the target site indicated by the black radial lines (see Sect. 2.3.2 for details). Interestingly, non-local ice-core locations systematically show the strongest relationship with the target site temperature.

However, after weighting the distance distribution with the average inverse number of available grid cells per distance bin, still only about one fifth of the optimal distances lie within 500 km from the targets.

## 3.4   Optimal ice-core sampling structures

The approach for choosing optimal ice-core locations yields straightforward and instructive results. However, it might be doubtful as to whether these results can be directly applied to the real world, since they might depend on the specific simulated climate state, the specific climate model and model isotope scheme used, or result from statistical overfitting. We therefore adapt our approach in a next step to learn more about the general spatial arrangement of the optimal ice-core locations which yield the maximum correlation with temperature. This is done by applying our concept of *sampling correlation structures* (see

Sect. 2.3.2 and the illustration in Fig. 1), which studies the correlation patterns only as a function of radial distance from the target site, averaging across 250 km radial bins and across the angular positions, thereby reducing local variability in the model data. Additionally, we apply the approach to all target sites in our DML and Vostok study regions (Sect. 2.3.3) and average the results across these sites to obtain regional estimates. Finally, we analyse each of the model variables to highlight the differences between the individual fields.

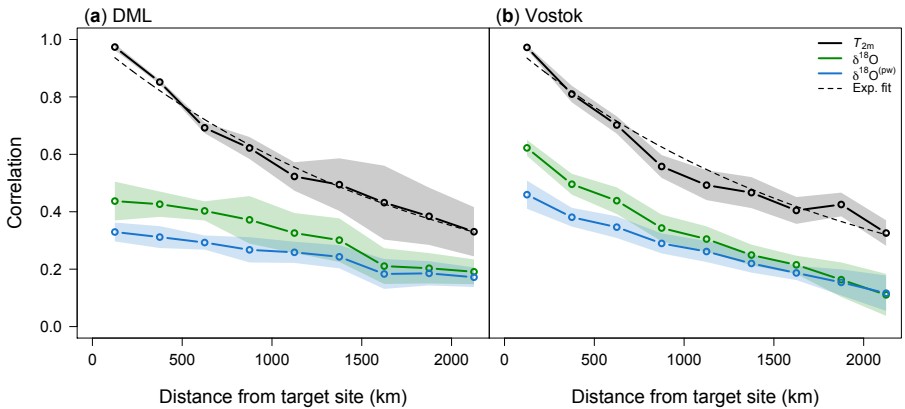

**Figure 5.** Average sampling correlation structures with temperature for the DML and Vostok regions in the case of sampling single locations. Shown is as a function of distance the average correlation between the interannual near-surface temperature ($T_{2\mathrm{m}}$) at a target site and the spatial fields of $T_{2\mathrm{m}}$ (black), oxygen isotope composition ($\delta^{18}$O, green) and precipitation-weighted oxygen isotope composition ($\delta^{18}$O$^{(\mathrm{pw})}$, blue). Averaging was performed in two steps: first, for a given target site, the correlations to the target site temperature were averaged across grid cells lying within 250 km wide consecutive rings around the given target site. Secondly, this analysis was conducted for all target sites in the DML (**a**) and Vostok (**b**) region and the results were aevarged across the respective region (see Sects. 2.3.2 and 2.3.3 for details). Shading denotes $\pm 1$ standard deviations of the correlation results across the different target sites in each region. The black dashed lines indicate an exponential fit to the $T_{2\mathrm{m}}$ data.

When we sample only a single location ($N_\ell = 1$), the sampling correlation structure is conceptually equivalent to the average correlation with distance, and it therefore simply gives the spatial decorrelation in the case of sampling from the $T_{2\mathrm{m}}$ field itself. The average sampling correlation structures for $T_{2\mathrm{m}}$ across the DML and Vostok regions (Fig. 5) can be described by an exponential decay with a length scale of $\sim 1900$ km in both cases, consistent with the estimated spatial temperature decorrelation lengths for the individual grid cells in these regions (Fig. 2a). In accordance with the general expectation, the

maximum average correlation with the target site temperature is thus obtained from sampling the innermost ring only.

     When we compare these results to the average sampling correlation structure for the $\delta^{18}$O field, we find for the DML region a much lower average correlation with the target site temperature as a function of distance (Fig. 5a). The average correlation for the innermost ring ($< 250$ km) is $\sim 0.4$, but decreases only slightly within the first $\sim 800$ km, followed by a little steeper decrease and near constant levels of $r \lesssim 0.2$ for distances $\gtrsim 1600$ km. For the Vostok region (Fig. 5b), the average sampling

correlation structure for $\delta^{18}$O exhibits a nearly linear decrease from an initial value of $r \sim 0.6$ to $r \sim 0.1$ in the final ring ($> 2000$ km). When we analyse the $\delta^{18}$O$^{(\mathrm{pw})}$ fields we find that precipitation weighting reduces the correlation values in both regions, but that it does not have a large effect on the shape of the sampling correlation structures itself.

     Extending this analysis to the two-dimensional case of sampling and averaging $N_\ell = 2$ locations offers the possibility of investigating the average correlation not only as a function of distance from the target site but implicitly also as a function of

distance between the two sampled locations (Fig. 6). The difference in the average sampling correlation structure between the

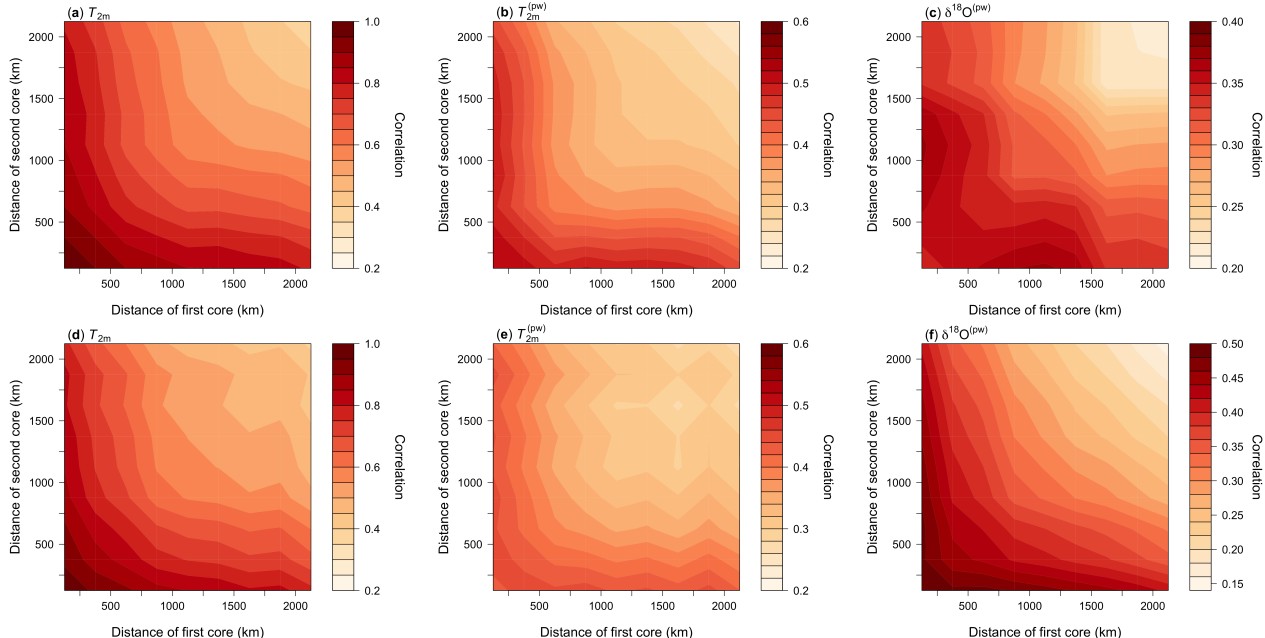

**Figure 6.** Average sampling correlation structure with temperature for the DML and Vostok regions in the two-dimensional case of sampling two locations. Shown is the mean correlation of all possible single correlations to the target site temperature of the average of two grid cells of (**a, d**) $T_{2m}$, (**b, e**) $T_{2m}^{(pw)}$ and (**c, f**) $\delta^{18}O^{(pw)}$ time series sampled from the same ring or from two different rings. This analysis was conducted for every target site in the DML region (panels **a–c**) and in the Vostok region (panels **d–f**) and the results were then averaged across the respective region. For each plot, the axes display the distance from the target site, where the $x$ ($y$) axis represents the first (second) sampled ring, with the results being mirrored along the diagonal for aesthetic reasons; the tick marks indicate the border distances of the rings. Note the marked difference in the locations of the correlation maxima between $T_{2m}$ and $\delta^{18}O^{(pw)}$ for the DML region, and also for the Vostok region the – albeit marginal – correlation maximum for $\delta^{18}O^{(pw)}$ is achieved by combining the innermost ring with the ring between 500–750 km.

fields of $T_{2m}$ and $\delta^{18}O^{(pw)}$ is even more pronounced for $N_\ell = 2$ than for $N_\ell = 1$. The maximum average correlation for $T_{2m}$ is still found when both sampling locations lie inside the innermost ring, as shown for the DML region (Fig. 6a). However, for $\delta^{18}O^{(pw)}$ the optimal arrangement of two locations is to sample one location from within the innermost ring but the second location from within the fifth ring, i.e., between $\sim 1000$ and $1250$ km from the target site (Fig. 6c). Part of this structure is

related to the effect of precipitation intermittency, which can be seen from the average sampling correlation structure of the $T_{2m}^{(pw)}$ field (Fig. 6b). Here, in contrast to $T_{2m}$, the correlation is about as high when we combine the innermost ring and one ring further away, as when we sample both locations from within the innermost ring.

Analysing the Vostok study region leads to comparable results (Fig. 6d–f), with a similar difference in average sampling correlation structure between $T_{2m}$ and $T_{2m}^{(pw)}$ as for the DML region, and a similar structure of $T_{2m}^{(pw)}$ and $\delta^{18}O^{(pw)}$ for distances

$\lesssim 1000$ km. However, the results for the $\delta^{18}O^{(pw)}$ field (Fig. 6f) do not display such a pronounced maximum correlation when

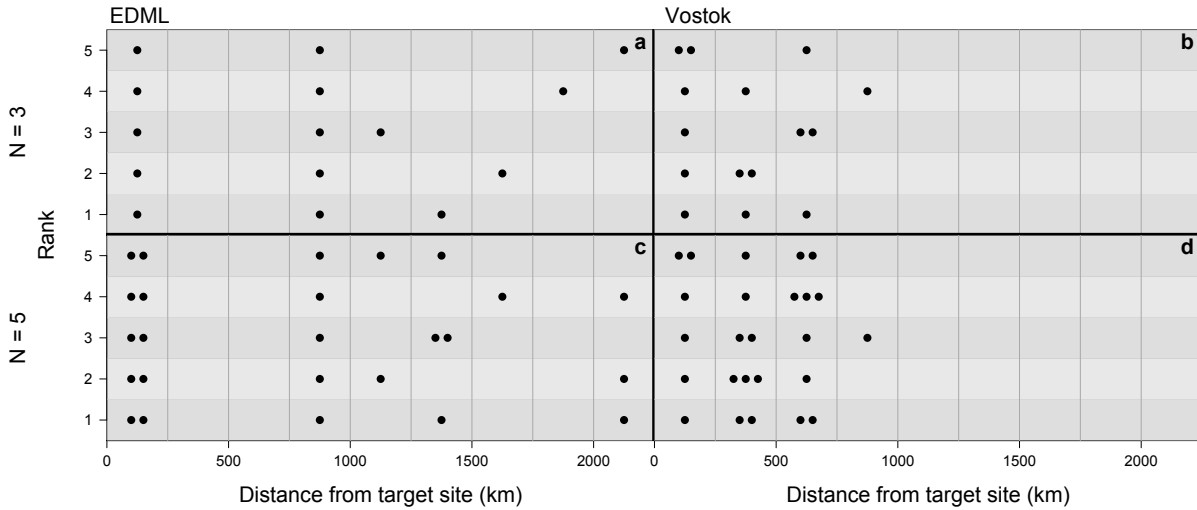

**Figure 7.** The optimal arrangement for averaging three or five $\delta^{18}O^{(pw)}$ ice cores to reconstruct the target site temperature at the EDML (**a**, **c**) and Vostok (**b**, **d**) drilling sites. Displayed are subsets of the sampling correlation structures for $N_\ell = 3$ and 5, showing along the vertical axis the optimal five of all possible combinations of rings (best denoted as rank 1, fifth best as rank 5), i.e., those which exhibit the five highest mean correlation values across $10^5$ random trials of averaging $N_\ell = 3$ (**a**, **b**) or $N_\ell = 5$ (**c**, **d**) grid cells from these rings. The ring bin borders are marked by thin vertical lines with their distances from the target site given on the horizontal axes; the selected optimal ring combinations are marked as black dots. Systematically, arrangements which combine ice cores from the innermost ring with ice cores further away are found to be optimal, with larger distances for the EDML target site.

one location is sampled from within the innermost ring and the second one from inside a ring further away as it is observed for the DML region. This suggests that the regional differences in the spatial correlation structure of the $\delta^{18}O$ field (Fig. 5) have an influence here.

The general feature of the optimal $\delta^{18}O^{(pw)}$ sampling arrangement is robust throughout Antarctica, despite the above re-
gional differences. When we analyse all available Antarctic target sites and fix the first core location to lie inside the innermost ring, in $\sim 82\,\%$ of all cases the optimal second core location is at least the second ring ($> 250\,\mathrm{km}$), and in $\sim 63\,\%$ of the cases it is the second to fourth ring ($250$–$1000\,\mathrm{km}$).

We obtain similar results also when averaging $N_\ell = 3$ or 5 locations of the $\delta^{18}O^{(pw)}$ field to reconstruct the target site temperature (Fig. 7). For computational reasons, we here only analyse single target sites. When EDML is set as the target site,
the optimal sampling configuration is such that 1–2 core locations lie in the innermost ring while the others are distributed at distances mostly between $\sim 750$ and $1500\,\mathrm{km}$ from the target. For reconstructing the Vostok target site temperature, the optimal core locations combine the innermost ring with locations distributed mostly across the second to third ($250$–$750\,\mathrm{km}$) ring.

In summary, averaging the $\delta^{18}O^{(pw)}$ time series across the optimal locations clearly increases the average correlation to the target site temperature more strongly with the number of locations, as compared to sampling all core locations only locally
close to the target site, i.e. from the grid cells that lie within the innermost ring (Fig. 8a). While the local correlation for the

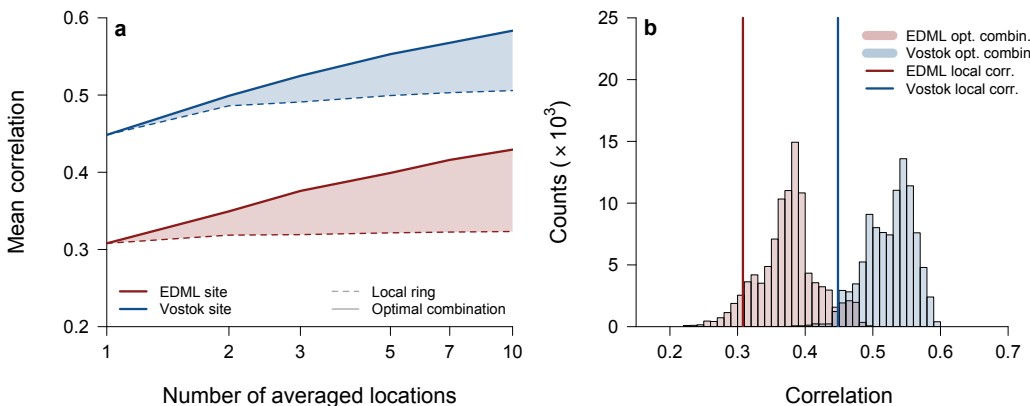

**Figure 8.** Gain in correlation and risk of adverse sampling. (**a**) The average correlation with the target temperature at the EDML (red) and Vostok (blue) sites depending on the number of locations, $N_\ell$, used for averaging the $\delta^{18}O^{(pw)}$ time series. Sampling is performed either locally from the innermost ring only (dashed lines), or from all possible individual combinations of locations for the respective optimal ring combination determined for each $N_\ell$ (solid lines). Compared to the local samples which show virtually no or only a small increase with the number of sampled locations, the correlation increases markedly with $N_\ell$ when sampling from the optimal rings, as highlighted by the shaded area. (**b**) Histogram of individual correlations for sampling from the optimal ring combination when averaging $N_\ell = 3$ locations compared to the correlation (vertical lines) for sampling from the innermost ring only, displayed for the EDML (red) and Vostok (blue) target sites. In both cases, the correlation is higher than the local value for more than $93\%$ of the optimal ring combination samples.

EDML target site stays constant around $0.31$, the optimal correlation rises to $0.35$ for $N_\ell = 2$ and to $0.43$ for $N_\ell = 10$, which is equivalent to nearly a doubling in the explained variance. For the Vostok target site, we observe a nearly concurrent increase in correlation between the local and the optimal sampling up until $N_\ell = 2$ from $0.45$ to $\sim 0.50$, but for larger $N_\ell$ the optimal correlation also increases more strongly and reaches $0.58$ for $N_\ell = 10$, a $\sim 1.7$-fold higher explained variance compared to $N_\ell = 1$.

These results are the mean value from averaging across many possible combinations of individual locations. In reality, any new drilling campaign or reanalysis of existing ice cores only represents one single combination of locations. Therefore, we further assess the risk of an "adverse optimal sampling", i.e., the probability of choosing by chance a specific sampling realisation from the optimal ring combination which yields a lower correlation than the correlation for sampling locally. For this purpose, we compare the distribution of individual correlations from sampling the optimal ring combination with the value obtained from sampling only the local sites which lie in the innermost ring. Overall we find the risk of adverse optimal sampling to be low, since more than $93\%$ of all individual correlation values in the example of $N_\ell = 3$ are actually larger than the respective local correlation (Fig. 8b).

## 4    Discussion

### 4.1    Dependence on radial distance

Oxygen isotope records derived from ice cores are commonly interpreted to reflect local temperature changes at the ice-core drilling site. Here we have shown that while there is local isotope–temperature correlation (Fig. 2b), this correlation can be increased considerably by averaging isotope records across space (Fig. 8a) following a distinct radial pattern which combines the local target site with locations located between a few hundred kilometres to up to $\sim 1000\,\text{km}$ from the target site (Figs. 6c, 6f, and 7). These results are based on a method which investigates the spatial correlation structure only as a function of radial distance by averaging across the azimuthal component. The motivation for this approach is that from physical arguments we expect the first-order spatial correlation patterns to be invariant against rotation. Such radial symmetry is indeed observed as the leading component of the spatial correlation structure of the temperature field and as a second-order component of the oxygen isotope field (Fig. 3). We interpret these symmetric contributions as a general feature of the underlying atmospheric processes, as compared to individual, local correlation maxima which are more due to the actual dynamics. Therefore, we expect that our results obtained from the radial sampling correlation structures should be largely independent of the climate state, or the climate model used, and thus serve as valid recommendations for real-world applications. In the next section, we substantiate this interpretation by showing that a simple conceptual model can predict the sampling correlation structure from the basic processes which shape the isotopic composition time series, modelled only as a function of radial distance. Finally, we will discuss the relevance of our results to actual ice-core studies.

### 4.2    Conceptual model of the optimal sampling structure

For a conceptual model of the sampling correlation structure, we focus on the three main atmospheric processes that influence the oxygen isotope records in ice cores: (i) temperature variations, (ii) precipitation intermittency, and (iii) the temperature–isotope relationship. We statistically model the associated fields of $T_{2\text{m}}$, $T_{2\text{m}}^{(\text{pw})}$ and $\delta^{18}\text{O}^{(\text{pw})}$ separately in order to understand the influence of each process (see Appendix A for details), and we assess, for comparable results, the predicted average sampling correlation structure with the target site temperature in the two-dimensional case of averaging two locations in the same manner that we analysed the climate model data.

To model the atmospheric temperature field, we assume an isotropic exponential decay of the spatial correlation with a constant decorrelation length (Appendix A2). Such an exponential temperature decorrelation is a commonly observed feature (Jones et al., 1997) and also confirmed by our climate model data (Figs. 2a, 3a, e, and 5). Given this relationship, we find a good agreement for the two-dimensional sampling correlation structure between the conceptual model and the climate model data, both regarding absolute correlation values as well as the spatial pattern (Fig. A2a). We emphasise that the maximum correlation with the target site temperature naturally occurs, in case of an isotropic correlation decay, when the averaged two (or $N_\ell$) locations are close to the target site, as any location which is further away will contribute a temperature signal that is less similar to the other locations.

To elucidate the role of precipitation intermittency, we follow the simplest assumption which is that this process can be described by partly aliasing the original temperature signal into temporal white noise (Laepple et al., 2018; Casado et al., 2020). We further assume that this noise is not independent between sites but that it follows the spatial scale of precipitation events, which we describe as an exponential decorrelation in space with a second length scale (Appendix A3). This intermittency length scale is related to the atmospheric processes that deliver precipitation, e.g., synoptic systems, and is hence assumed to be smaller than the length scale of the temperature anomalies. The introduction of this second length scale into our conceptual model generally explains the optimal sampling structure we obtained from the climate model data. Qualitatively, close-by locations exhibit a strong correlation in temperature but also in the noise from precipitation intermittency; therefore, this noise cannot be reduced by averaging the locations, yielding an overall low signal-to-noise ratio. However, with increasing distance between the locations, the intermittency noise decorrelates faster than the temperature field due to the different decorrelation scales, resulting in an optimal distance of maximum signal-to-noise ratio. This is also reflected in our conceptual model (Figs. A1 and A2b, e): When fixing the position of one core to the innermost location and varying only the distance from the target site of the second core location, the correlation with the target site temperature first increases with increasing distance of the second location and then maximises at an optimal distance, before it decays with a further increase in distance. In the climate model data, we observed a similar feature for the precipitation-weighted temperature (Fig. 6), though it was not as clear as in the conceptual model. This mismatch could be related to the assumed isotropy in the conceptual model and the according azimuthal averaging done in the climate model data analysis, which potentially smears the intermittency effect in the climate model data due to slight differences in the decorrelation lengths between the different horizontal directions.

In order to incorporate the $\delta^{18}O^{(\text{pw})}$ field into the conceptual model, we need to account for the spatial temperature–isotope relationship. To accomplish this, we parameterise the spatial dependence of the correlation between temperature and the oxygen isotope composition with a simple isotropic linear model based on the climate model data results (Fig. 5 and Appendix A4). In addition, we assume that the same effect of precipitation intermittency that we adopted for the temperature field is also applicable to the oxygen isotope field. With these simple assumptions, we obtain a good qualitative agreement for the DML region between the conceptual model and the climate model data results (cf. Figs. A2c and 6c). In addition, when we change the parameterised isotope–temperature relationship such that it more closely resembles the Vostok region data (Fig. 5b), the sampling correlation structure in the conceptual model (Fig. A2f) is more similar to the observed correlation structure (Fig. 6f). However, in general the conceptual model fails for $\delta^{18}O^{(\text{pw})}$ to reproduce the actual range in correlations as it produces much lower values than expected.

In summary, our conceptual model provides a quantitative understanding of the spatial correlation of the temperature in the climate model data, and, at least, a qualitative understanding of the processes that affect the correlation between temperature and the $\delta^{18}O^{(\text{pw})}$ field, i.e., precipitation intermittency and the spatial temperature–isotope relationship. The deficiencies in the conceptual model may be attributed to its simplicity. For the governing processes, we assumed spatially constant and isotropic length scales, neglecting local and direction-related differences in, e.g., temperature decorrelation lengths (cf. Fig. 2a) or the spatial extent of the coherence of precipitation intermittency. Instead of being constant, the latter may differ depending on the type of precipitation, e.g., synoptic versus stratiform precipitation, and may exhibit directional dependencies related to

topography. Furthermore, we assumed constant variance of all time series, thereby ignoring potential weighting effects on the correlations for the spatial average of several locations due to different variabilities between them.

## 4.3 Relevance for ice-core studies

Our results from analysing the climate model data provide guidance on where to drill or from where to analyse $N_\ell = 1, 2, 3$
or more ice cores in order to optimally reconstruct the atmospheric temperature signal for a certain target site. For this, our analysis highlights two distinct approaches.

The first possibility is to follow the recommendations obtained from directly choosing the specific locations which maximise the correlation with the target site temperature (Fig. 4). This is straightforward, however, for applications such locations would need to be derived for every target site in question. In addition, as outlined above, it is unclear whether the results can be
one-to-one transferred to the real world, since they might be due to unaccounted model deficiencies, or depend on dynamical processes in the atmosphere which could differ between climate states or depend on initial conditions. One indication for this is that we obtain different optimal single core locations for more than half of all investigated Antarctic target sites, when we analyse only the first or only the second half of the respective climate model time series as compared to the full 1200 years.

We have argued above that the sampling correlation structures, obtained from averaging the individual correlations across
space for combinations of concentric rings around the target site, are a more general quantity, and we have shown with our conceptual model that they are on average governed by the interplay of the different underlying correlation length scales. We expect the latter to vary less in between different climate periods or states, or in between regions. This is substantiated by the fact that the sampling correlation structures for two cores (Fig. 6) are much more robust against analysing only the first or the second half of the model time series, different to the results from directly choosing optimal locations. Thus, the sampling
correlation structure offers a general approach for finding an optimal ice core network, but with the downside that it informs us only about the relative radial distances of the optimal network around the target site.

Using the sampling correlation structures we arrive at the following recommendations for optimal ice core sampling configurations. If it is only possible to drill or analyse a single ice core, our results show that it is always best to sample locally, i.e., to place this core near the target site of interest. This is also common practice, given the usual interpretation of ice-core isotope
records as a proxy for local temperatures. However, due to the effect of precipitation intermittency, modulated by the spatial coherence of the temperature–isotope relationship, it is no longer optimal in the case of drilling two ice cores to collect both cores near the target site, but instead to drill one core at the target site and one at least $500\,\mathrm{km}$ away. Where three or more ice cores will be drilled or analysed, we expect the optimal spatial configuration to be more dependent on the study region, but our results indicate that it is still likely best to place one core near the target site and distribute the others across several hundreds
of kilometres.

These inferences are based on data from a single climate model simulation together with a simple statistical conceptual model, which should be tested against observations. As a proof of concept, we thus need to create an isotope record that is in first order only governed by temperature variations and precipitation intermittency, and remove the impact of small-scale stratigraphic noise from the actual measured records (assuming that any further processes in the pre-depositional to depositional

phase contribute negligibly to the local isotopic variations). To accomplish this, one possible strategy would be to use trench sampling campaigns (see Münch et al., 2016, 2017 for the EDML site). Then, one test of our optimal sampling configurations could be to combine one trench record, e.g., one from EDML, with another trench sampled at the optimal distance based on our results for $N_\ell = 2$, and correlate the average of these two trench records with the instrumental temperature data set available for EDML. Based on the results in this study we would expect a higher degree of correlation in this case compared to using only

one local trench record from EDML. We acknowledge that such an approach would be challenging due to the small amount of available instrumental data ($\sim 20$ years for EDML) and by the inevitable dating uncertainties between the two trench records.

Finally, we note that our implications concerning optimal ice core sampling configurations might in reality be affected by two further processes we have neglected here. Firstly, clear sky precipitation ("diamond dust") is a common phenomenon in Antarctica, especially in the drier regions of the Antarctic Plateau, which potentially occurs more regular than convective-type

or stratiform precipitation. Diamond dust formation is not explicitly simulated by the ECHAM5 model, so it is possible that the precipitation intermittency modelled in our simulation is partly offset in reality by a stronger relative contribution of diamond dust to the total precipitation amount. Secondly, surface–atmosphere vapour exchange in between precipitation events might constitute a second process which imprints an atmospheric temperature signal into the surface snow, next to precipitation (e.g., Steen-Larsen et al., 2014; Madsen et al., 2019). This process could hence also partly counteract the impact of precipitation

intermittency, depending on its relative importance for the isotopic composition of the surface snow. However, there is no clear consensus in the recent literature on this question, and ultimately we need quantitative estimates of the importance of vapour exchange processes across temporal scales. In any case, these considerations do not affect our general notion that the optimal ice core sampling configuration depends on the differences in spatial decorrelation scales of the processes which shape the isotopic records.

**5 Conclusions**

In this study we assessed the spatial sampling configuration of ice cores to optimally reconstruct the annual near-surface temperature at a specific target site. This problem was motivated by the expectation that the major processes influencing the isotopic records of ice cores operate on different spatial scales.

Indeed, by analysing the temperature and isotope data of an isotope-enabled atmosphere–ocean climate model simulating the

climatic history over the last millennium in Antarctica, we showed that while in the optimal setup a single ice core should be placed close to the target site of interest, a second core should be located far ($> 500$ km) from the first core. While this may seem surprising at first glance, it can be straightforwardly explained by the interplay of two different correlation lengths in space: one for the temperature anomalies and one parameterising the spatial coherence of the effect of precipitation intermittency, as demonstrated by a simple conceptual model. Despite the fact that these results were specifically obtained for two regions of the

East Antarctic Plateau, we expect similar results to hold for other parts of Antarctica, and potentially also for other large-scale ice-coring regions such as Greenland, as long as our simplified assumptions of nearly isotropic exponential decorrelation length

scales are also valid there. However, we also suggest to verify our results with a different isotope-enabled climate model in order to rule out any dependence on the specific atmospheric model and isotope scheme applied in the simulation used here.

Overall, our study explicitly improves the planning of drilling or analysis campaigns for spatial networks of ice-core isotope
records. In addition, it provides a strategy to analyse an optimal configuration of sampling locations for any proxy which is influenced by two or more processes that exhibit different spatial correlation scales. This likely applies to various marine as well as terrestrial proxy types, and our strategy thus might offer a step forward in the best use of sampling and measurement capacity for quantitative climate reconstructions, which needs to be investigated in further studies.

*Code and data availability.* The climate model data used in this study is freely available from the Zenodo database under https://doi.org/10.
5281/zenodo.4001565 (Münch and Werner, 2020). Software to run the analyses and produce the figures is available as R code hosted in the public git repository at https://github.com/EarthSystemDiagnostics/optimalcores; a snapshot of this software code at the time of publication is archived on the Zenodo database under https://doi.org/10.5281/zenodo.5075439 (Münch, 2021).

## Appendix A: Conceptual model of sampling correlation structures

### A1 General model

We set up a conceptual model for the correlation between a target temperature time series and a spatial average based on a set of locations sampled from a climatic field (sampling correlation structure). Our model assumes simple isotropic and exponential decorrelation structures for the involved climatic fields and is based on previous work which suggests that precipitation intermittency can be described by partly aliasing the original temperature signal into white noise (Laepple et al., 2018).

In the model, we consider a temperature time series $T_0$ at some target site $\mathbf{r}_0$ and a field $x$ of a given climate variable. From
this field, we select $N_\ell$ time series $x_i$ at the locations $\mathbf{r}_i$, $i = 1, \ldots, N_\ell$, and denote the spatial average of these time series by $\overline{x} = \frac{1}{N_\ell} \sum_{i=1}^{N_\ell} x_i$. The distances of the $N_\ell$ locations from the target site and the distances between the locations are given by $r_i = |\mathbf{r}_i - \mathbf{r}_0|$ and by $d_{ij} = |\mathbf{r}_i - \mathbf{r}_j|$, respectively. The correlation between $T_0$ and $\overline{x}$ follows from

$$\mathrm{cor}(T_0, \overline{x}) = \frac{\mathrm{cov}(T_0, \overline{x})}{\sqrt{\mathrm{var}(T_0)\mathrm{var}(\overline{x})}}, \tag{A1}$$

and it is governed by the covariance between the temperature at the target site and the climate field at the sampling locations
$\mathbf{r}_i$,

$$\mathrm{cov}(T_0, \overline{x}) = \frac{1}{N_\ell} \sum_i^{N_\ell} \mathrm{cov}(T_0, x_i), \tag{A2}$$

and by the covariance between the sampling locations through the variance of their spatial average,

$$\mathrm{var}(\overline{x}) = \frac{1}{N_\ell^2} \left( \sum_i^{N_\ell} \mathrm{var}(x_i) + 2 \sum_i^{N_\ell - 1} \sum_j^{N_\ell} \mathrm{cov}(x_i, x_j) \right). \tag{A3}$$

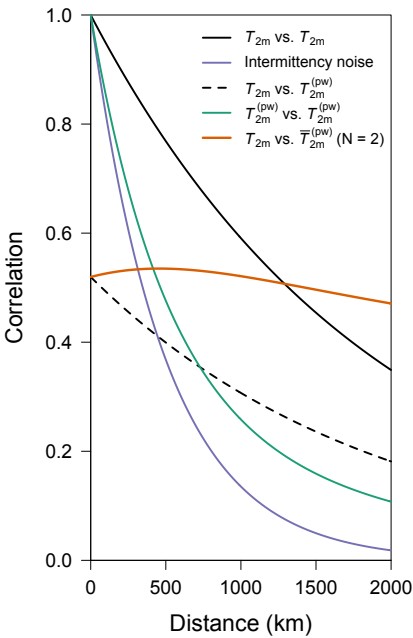

**Figure A1.** Illustration of the decorrelation lengths in the conceptual model. Shown are as a function of distance the correlation between two temperature time series (black), between the intermittency noise (purple), between a temperature and a precipitation-weighted temperature time series (dashed black), between two precipitation-weighted temperature time series (green), and between a target temperature time series and the average of two precipitation-weighted temperature time series (orange), when one is located at the target site and the other one is located away from the target site at a distance as indicated on the x axis. Model parameters are taken from the DML region. The decorrelation curve of the precipitation-weighted temperature time series is simply the superposition of the temperature decorrelation and the decorrelation of the intermittency noise, depending on the intermittency factor $\xi$.

In our model, these quantities depend on the distance between sites and on the correlation structure of the respective field $x$, as we show in the following and as illustrated in Fig. A1.

## A2 Temperature

For the near-surface temperature field, $x \equiv T$, we assume a spatially constant variance, $\mathrm{var}(T_0) = \mathrm{var}(T_i) \equiv \sigma_T^2$, and an isotropic decorrelation following an exponential decay with a decorrelation length $\tau$; i.e., the covariance between sites is (see black line in Fig. A1)

$$\mathrm{cov}(T_0, T_i) = \sigma_T^2 \exp\left(-\frac{r_i}{\tau}\right), \tag{A4}$$

$$\mathrm{cov}(T_i, T_j) = \sigma_T^2 \exp\left(-\frac{d_{ij}}{\tau}\right). \tag{A5}$$

The correlation between the target site temperature and the spatial average of $N_\ell$ temperature time series is then obtained from

$$\mathrm{cor}(T_0, \overline{T}) = \frac{\sum_{i=1}^{N_\ell} \exp\left(-\frac{r_i}{\tau}\right)}{\sqrt{N_\ell + 2\sum_{i=1}^{N_\ell-1} \sum_{j=i+1}^{N_\ell} \exp\left(-\frac{d_{ij}}{\tau}\right)}}. \tag{A6}$$

## A3   Precipitation-weighted temperature

To model the effect of precipitation intermittency, we follow Laepple et al. (2018) and assume that precipitation intermittency redistributes the energy of the temperature time series constantly across frequencies, i.e., creating temporal white noise without changing the total variance. Then, the precipitation-weighted temperature time series at location $\mathbf{r}_i$ arises from $T_i$ as

$$T_i^{(\mathrm{pw})} = (1-\xi)^{1/2} T_i + \xi^{1/2} \sigma_T \varepsilon_i(0,1), \tag{A7}$$

where $\varepsilon_i(0,1)$ are independent and normally distributed random variables with a mean of zero and a standard deviation of 1. The parameter $0 \le \xi \le 1$ determines the fraction of the input temperature time series which is aliased into white noise.

The covariance between the target site temperature and a precipitation-weighted temperature time series is then

$$\mathrm{cov}(T_0, T_i^{(\mathrm{pw})}) = (1-\xi)^{1/2} \sigma_T^2 \exp\left(-\frac{r_i}{\tau}\right), \tag{A8}$$

which implies that the spatial correlation structure between $T_0$ and the precipitation-weighted temperature follows the same exponential decay as in Eq. (A4), only scaled by the factor $(1-\xi)^{1/2}$ (see dashed black line in Fig. A1). The factor $\xi$ can be estimated from the climate model data by analysing the local correlation, i.e., at the same grid cell, between the temperature and the precipitation-weighted temperature.

We further assume that the effect of precipitation intermittency is not independent between sites but is related to the spatial coherence of the precipitation fields, for which we assume an exponential decorrelation structure with a decay length $\tau_{\mathrm{pw}}$. Based on these assumptions, the spatial covariance between sites of the white noise terms induced by the effect of precipitation intermittency has the form (see purple line in Fig. A1)

$$\mathrm{cov}(\varepsilon_i, \varepsilon_j) = \exp\left(-\frac{d_{ij}}{\tau_{\mathrm{pw}}}\right). \tag{A9}$$

Then, the correlation between the target site temperature and the spatial average of $N_\ell$ precipitation-weighted temperature time series is governed by the intermittency factor $\xi$ and by the two spatial length scales $\tau$ and $\tau_{\mathrm{pw}}$,

$$\mathrm{cor}\left(T_0, \overline{T}^{(\mathrm{pw})}\right) = \frac{\sqrt{1-\xi}\sum_{i=1}^{N_\ell} \exp\left(-\frac{r_i}{\tau}\right)}{\sqrt{N_\ell + 2\sum_{i=1}^{N_\ell-1} \sum_{j=i+1}^{N_\ell} g(d_{ij}; \tau, \tau_{\mathrm{pw}}, \xi)}}, \tag{A10}$$

with

$$g(d_{ij}; \tau, \tau_{\mathrm{pw}}, \xi) := (1-\xi)\exp\left(-\frac{d_{ij}}{\tau}\right) + \xi\exp\left(-\frac{d_{ij}}{\tau_{\mathrm{pw}}}\right). \tag{A11}$$

An example of the correlation according to Eq. (A10) is shown for $N_\ell = 2$ and $r_1 = 0$ as a function of $r_2$ in Fig. A1.

## A4 Precipitation-weighted oxygen isotope composition

For the precipitation-weighted oxygen isotope composition field, $x \equiv \delta^{(\mathrm{pw})}$, we assume the same effect of precipitation intermittency as for the temperature field. Furthermore, an analysis of the climate model data suggests that the oxygen isotope field largely exhibits an exponential decorrelation structure in space (not shown). Hence, the correlation between the target site temperature and the spatial average of $N_\ell$ $\delta^{(\mathrm{pw})}$ time series is obtained in a similar manner as for $T^{(\mathrm{pw})}$, i.e.,

$$\mathrm{cor}\left(T_0, \overline{\delta}^{(\mathrm{pw})}\right) = \frac{\sqrt{1-\xi}\sum_{i=1}^{N_\ell}\mathrm{cor}\left(T_0, \delta_i\right)}{\sqrt{N_\ell + 2\sum_{i=1}^{N_\ell-1}\sum_{j=i+1}^{N_\ell}g(d_{ij};\tau_\delta,\tau_{\mathrm{pw}},\xi)}}, \tag{A12}$$

where $\tau_\delta$ is the decorrelation length of the $\delta$ field and the only difference to Eq. (A10) is the unknown spatial correlation structure between the temperature at the target site and the oxygen isotope field, $\mathrm{cor}\left(T_0, \delta_i\right)$. Based on our climate model results (Fig. 5), we parameterise this function with a simple linear decay of the form

$$\mathrm{cor}\left(T_0, \delta_i\right) = \begin{cases} c_0 - \gamma d, & d \le d_0, \\ 0, & d > d_0, \end{cases} \tag{A13}$$

where $\gamma = c_0/d_0$, and $d_0$ is some threshold distance above which the correlation is zero.

## A5 Model parameter estimation and model results

Overall, our model is governed by three decorrelation lengths ($\tau$, $\tau_\delta$, $\tau_{\mathrm{pw}}$), the intermittency factor $\xi$, and two parameters describing the temperature–isotope correlation ($c_0$, $d_0$).

We estimate $\tau$ from the climate model data for the DML and Vostok regions (Fig. 5) and find for both regions values of $\tau = 1900\,\mathrm{km}$. In the same way we estimate a value of $\tau_\delta = 1100\,\mathrm{km}$ for both regions. The intermittency factor $\xi$ is derived from the local correlation between temperature and precipitation-weighted temperature (Eq. A8). We find an average value for the DML region of $\xi_{\mathrm{DML}} = 0.73$, which is close to the average value across all of Antarctica ($\xi_{\mathrm{Ant.}} = 0.71$), while the intermittency is stronger for the Vostok region ($\xi_{\mathrm{Vostok}} = 0.82$). We parameterise the temperature–isotope correlation in the DML region with $c_0 = 0.4$ and $d_0 = 6000\,\mathrm{km}$ and in the Vostok region with $c_0 = 0.6$ and $d_0 = 2500\,\mathrm{km}$ (Fig. 5). The only unconstrained parameter is the decorrelation length of the effect of precipitation intermittency, $\tau_{\mathrm{pw}}$, since it is unclear by which precipitation variable it is mainly governed (total annual amount, seasonal amount, or its distribution). An investigation with reanalysis data yielded scales between $\sim 300$ to $500\,\mathrm{km}$ for different precipitation variables (Münch and Laepple, 2018), while our model data exhibits an average decorrelation length of $\sim 600\,\mathrm{km}$ for the annual precipitation amount. Here, for the conceptual model we choose a value of $500\,\mathrm{km}$.

We can test our assumption for the effect of intermittency based on using the estimated values of $\tau$ and $\xi$ to predict the spatial decorrelation between temperature and precipitation-weighted temperature (Eq. A8). Indeed, this yields a comparably good fit to the data as an independent fit (root mean square deviation of $\sim 0.03$ between data and fit in both cases), supporting our assumption that intermittency can be parameterised by a partial conversion of the time series into white noise.

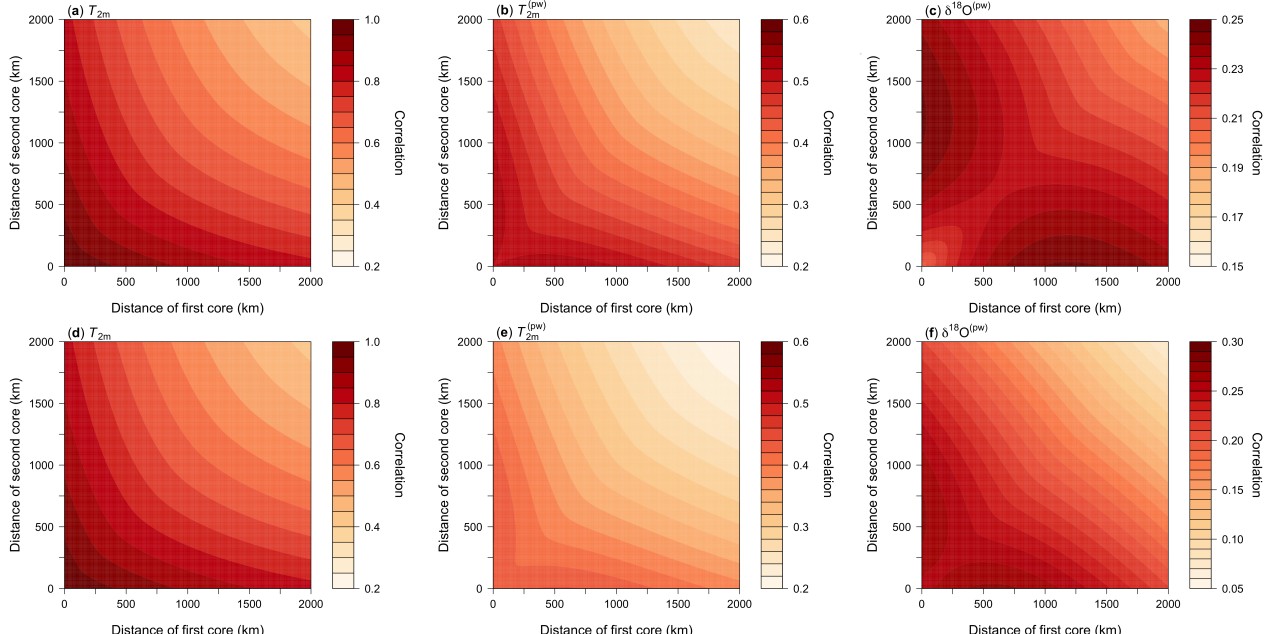

**Figure A2.** Two-dimensional sampling correlation structures with temperature as predicted from our conceptual model using the model parameters from the DML (**a–c**) and Vostok (**d–f**) regions. Shown is the mean correlation of all possible single correlations for the average of two time series sampled from a pair of concentric rings around the target site for the fields of (**a, d**) $T_{2m}$, (**b, e**) $T_{2m}^{(pw)}$ and (**c, f**) $\delta^{18}O^{(pw)}$. Note that the plots (**a**) and (**d**) are based on the same parameters and therefore identical.

Similarly to analysing the climate model data, we now use our conceptual model to predict the two-dimensional ($N_\ell = 2$) sampling correlation structures for the different model fields of $T_{2m}$, $T_{2m}^{(pw)}$ and $\delta^{18}O^{(pw)}$ (Eqs. A6, A10 and A12). Since

525 our model space is continuous, we sample from locations placed *on* concentric rings around the target site. We either sample the two locations from the same ring or from two different rings, using ring radii from 0 to 2000 km in increments of 10 km, and calculate the average correlation for a specific ring combination. To obtain meaningful expectation values, we choose 36 locations distributed uniformly across each ring in steps of $10°$, combine these locations one by one for each ring combination, and average across the correlations for each location pair. With the model parameters from the DML and Vostok regions we

530 obtain the results displayed in Fig. (A2), which are discussed and compared to the estimated results from the climate model data in the main text.

*Author contributions.* TM and TL designed the research and developed the methodology. MW contributed with providing the climate model data and with his expertise on the modelling of precipitation isotopic composition. TM processed the model data, coded the analysis software, performed the analyses and wrote the first version of the manuscript. All authors contributed to the interpretation of the results and to the

535 preparation and revision of the final manuscript.

*Competing interests.* The authors declare that they have no conflict of interest.

*Acknowledgements.* We thank Jesper Sjolte (Lund University) for performing the ECHAM5/MPI-OM-wiso past1000 model simulation, and Mathieu Casado, Raphaël Hébert and Torben Kunz (AWI) for their helpful comments on this project and the manuscript. All plots and numerical analyses in this paper were carried out using the open-source software R: A Language and Environment for Statistical Computing. We are grateful to the two reviewers Lenneke Jong and Dmitry Divine, and to István Hatvani and Zoltán Kern, whose comments helped to significantly improve the first version of this paper. Finally we thank Nerilie Abram for editing the paper.

540

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
