# Peer review of "How precipitation intermittency sets an optimal sampling distance for temperature reconstructions from Antarctic ice cores"

_Climate of the Past, 2020_

## Referee Comment (RC1) · Lenneke Jong (Referee) · 6 Nov 2020

**1 General Comments**

This manuscript investigates the possibility of selecting an optimal set of ice core locations to best reconstruct the temperature record at a target Antarctic site. They use an isotope enabled climate model to derive what they term a "sampling correlation structure" of sampling locations from concentric rings radiating out from a target site. Their findings that the optimal reconstruction can be obtained by combining a local core with another 500-1000km away to decrease the noise in the records due to precipitation

intermittency is particularly interesting.

The paper itself is quite well structured, however, I think it would benefit from including the details of the conceptual model earlier into the main text of the paper when the decorrelation length scales relating to the sources of noise are introduced. A diagram would also be of great help. It is probably beyond the scope of this paper but I would have liked the authors to comment on the relative importance of the different sources of noise in the isotopic signal and which ones are dominant depending on location in Antarctica (eg coastal vs inland).

I found some of the discussions of study regions and target regions quite confusing and could do with some clearer explanations. I also thought a motivating example with actual ice core data at one of the target sites to demonstrate the reduction in noise would be very helpful, and would be useful in quantifying how much improvement in the temperature reconstruction is obtained with additional cores when all the other confounding factors are included in the isotope signal. They state that including these sources of noise, eg isotope diffusion is outside the scope of this study, but and I would like to see how the results hold up when the real data is included.

The authors are clear in the assumptions going in to the conceptual model and the analysis overall. They are also clear on the limitations of trying to use this optimal sampling correlation structure in the real world.

Overall, the manuscript is written clearly, and is of good quality and is of scientific interest, especially to the ice coring an palaeoclimate community. I have listed a few specific comments below that I believe would help improve the readability of the paper and recommend publication once these issues are addressed.

**2 Specific Comments**

Lines 13-14: Remove final sentence as I didn't see the application of this technique to anything other than temperature reconstructions from ice cores discussed.

section 2.3.3: Some motivation and justification for the sampling scheme of rings and selection would be helpful. It was also not clear at times if $N$ was referring to the number of grid cells, or rings and also is presumably different to $N_{grid}$ mentioned later on line 32 onwards.

section 3.1 I would have liked to see what level of significance is attached to each correlation coefficient reported.

Fig 2: (And other figures) Use of diverging colour map for only positively increasing correlation or lengths scales is a bit distracting. Consider only using reds for instance.

Fig 3. This is an important figure in that it shows the precipitation weighting being important in the correlation. I would like to see some indication of where these correlations are significant to (eg p<0.05). It would be nice to see another map explicitly showing the difference in correlation coefficients between the two, as it looks to be some regions where there is no difference at all.

3.3 Optimal ice-core sampling structures line 187 "we compute the mean of correlation results obtained between a target site temperature and individual grid cells in order to reduce local variability in the model data" Is this what you really mean, averaging the correlations? This seems like a strange thing to do if you are looking to maximise the correlation overall?

I would also like more comment on how much of the regional difference is lost by averaging to only get a function of radial distance. As the DML and Vostok results differ and you suggest there are regional differences. Surely the correlations are not radially uniform?

Fig 4. It would be also useful to have the concentric circles marked around the target sites. Can you also comment on why the locations shift when more cores locations are added? That is, is the location in the N=1 case also included in the N=3 case as it looks as though they have moved slightly. These segments don't seem to correspond to the regions mapped out by the black polygons in Fig. 1 so it is not clear to me what the study regions actually are. Does it mean that the cores in the N>1 cases can be from outside of those black polygons too as appears to be the case here?

Fig 5. The black dashed line indicating the exponential fit is not included in the figure legend, only the caption. The dots in the plot don't appear to be at the expected 0, 250, 500km marks, but are a bit offset, is this deliberate? Again, I question the averaging step around the whole 250km rings, but it would be nice to see the spread in the correlation as a function of distance too. I am confused at what is meant by "all respective target sites in the DML and Vostok region", how are there more than one target sites for each region, are these different that the two crosses shown in Fig. 4?

Fig 6. These are interesting and show the clear difference for the precipitation weighted $\delta^{18}$O and $T_{2m}$ for DML, but why is the Vostok case relegated to the appendix? The fact that they are different is an interesting result.

Fig 7. The way this figure is arranged, is the top row, marked rank 5, that which has the max correlation, or is the rank 1 row the highest correlation row? I find it very curious that in the Vostok region the optimal arrangement comes with no local sites in many of the cases

line 220: Suggests that regional differences play a part here and I would like a comment on what those differences are (elevation, distance from coast etc?)

Line 232: Does the averaging have an effect on the significance of the correlations?

Fig. 8 (b) I assume the colours on the histogram are the same as in (a), but please add the legend anyway. Can you comment on the low correlation outliers for the EDML

case.

Section 4.1 I found most of the discussion very clear and thorough, but re-iterate that a good schematic diagram to illustrate the length scales in the conceptual model would be very useful.

---

## Short Comment (SC1) · 25 Nov 2020

Dear Authors,

First of all let us congratulate you on this very concise and precisely documented study which was highly interesting to read.

In the following we would like to comment on two key results.

We agree that ambient temperature of precipitation events should be expected to show a stronger correlation to precipitation d18O than annual mean temperature due to the intermittency of precipitation. We also agree that

the data calculated from the simulation results reflect this theoretical relationship, however we would like to note that in a more experimental approach (http://journals.pan.pl/dlibra/publication/116059/edition/100870/content) we tested this idea and an opposite result was obtained. We found that amount weighting is incapable of ameliorating the signal replication between the stations and the ice cores, while arithmetic means gave the stronger linear relationships. The explanation is thought to be isotopic exchange between vapor and surface snow. In the present paper this may open an additional perspective from which the contrast seen in Figure 3 and Sects. 3.1 and 4 can be viewed from.

We also agree with the concept that the signal can be enhanced by averaging isotope records across space, however it is quite strange that ". . .the optimal sampling strategy is to combine a local ice core with a more distant coreâĹij500–1000km away. A similarly large distance between cores is also optimal for reconstructions that average more than two isotope records." In this paper http://dx.doi.org/10.1016/j.polar.2017.04.001 we performed geostatistical analysis of 60 ice core derived d18O time series in Antarctica to determine their spatial autocorrelation structure and to find the area yet unrepresented by the assessed set of records. The spatial autocorrelation (varography; Matheron 1963) is not equivalent to decorrelation (Appendix B1-2) but also measures the spatial similarity of the studied parameter. For instance, we obtained a 350km spatial "influence" range of the assessed ice core d18O records via semivariogram analysis, which would be interesting to be compared with your results regarding the question: Why are the original ice core d18O data spatially correlated within 250km and the modeled ones in your study above 500 km to simplify the question...

These experimental findings based on real life data might worth consideration when your model results are evaluated and may also serve as a good addition to the discussion.

Best regards, Zoltán Kern & István Hatvani

---

## Referee Comment (RC2) · Dmitry Divine (Referee) · 2 Dec 2020

**Review of a manuscript for** *Climate of the Past*

**How precipitation intermittency sets an optimal sampling distance
for temperature reconstructions from Antarctic ice cores** by T. Munch et al.

**Overall:**

In this manuscript the authors present a theoretical/modelling framework for establishing the optimal spatial network of ice cores that maximizes the correlation of the derived composite annual series with the target air temperature series. The authors use the output of the past millennium ECHAM5/MPIOM simulation with isotopic tracers as the only data source for their study. Using Antarctic as a target region, they demonstrate that in the framework of the considered model, the optimal ice core network for reconstructing the temperature in a target location should cover the area of 500-1000 km to minimize the noise effects of precipitation intermittence on the T-d18O relationship.

The authors made a number of serious simplifying assumptions in their approach/analysis, such as anisotropic decorrelation scale, linear dependence of d18O in precipitation on condensation temperature and a condensation temperature on 2m air temperature, etc. However, these limitations are clearly presented in the text.

The paper is generally clearly written and results are well presented. I therefore consider the manuscript deserves to be published after some moderate modifications according to the comments provided below.

**Major comments:**

My major comment concerns the presentation of the sampling procedure in 2.3.2-2.3.3. which I have found not very straightforward to comprehend. One should admit I have spent quite some time trying to understand the actual details behind the technique, though this difficulty could of course be quite individual. The grip of understanding came later, while reading "Results", yet some questions still remain. A number of minor questions that emerged while reading the manuscript, could therefore be a result of my unclear understanding of the basics of the proposed method.

I would like to note also that sometimes the discussion around/use of terms like "target site" or "local site" may appear confusing, same as the actual dimension of the core network being discussed. May be some simplification/clarification of 2.3. can improve the readability?

In general, with respect to the sampling strategy, the question is why the authors initiated the procedure with these concentric rings used for spatial sampling, instead of just random seeding of the "sampling locations", calculating the metrics of interest and then ordering them according to the distances between the locations? It sounds way more straightforward to comprehend than via introducing these circular sampling areas with an increment of an arbitrary choice.

Also, as a suggestion for the future work, it would be highly useful to test the concept of this method on a different model with enabled stable water isotopes in precipitation in order to see how different the results/inference can be. Testing on the existing ice core network can be fairly problematic due to all the deficiencies (both in the available ice core and instrumental data) mentioned throughout the text.

**Minor comments (the manuscript text shown italics)**
Line 112: "*...define consecutive rings around this site with a 250 km radial width...*"
Here you refer to these concentric rings with a radius increment of 250 km, used for delimitation of the sampling regions, do I get it right? May be it needs to be specified already here. Can you also provide any rationale behind the value of 250 km?

Line 118: "*Finally, we report the mean correlation for every ring combination by averaging across all correlations of the analysed grid-cell combinations.*" Is this averaging based on the distance between the locations, or just everything? How then the distance-based value is calculated?

Line 119: "…for sampling N locations from the model field depending on the distances between the locations." See my previous comment. If everything is averaged out, how the distance based sorting/ranking is implemented?

Line 136: *"(−78.47 S)*». No need in "-" before the latitude value if "S" is explicitly indicated.

Line 184: "*...depend on the specific simulated climate state or result...*"
It would be meaningful to add that it also includes the actual model used and the stable water isotope scheme applied in the model

Line 212: "*...maximum average correlation is to sample one location from the innermost ring and the second location from the fifth ring*"
this is not entirely apparent as both maxima in Fig 6 seem to be found on the "5th ring".

Line 255:" For a conceptual model of the sampling correlation structure, we focus on three processes that influence…". It is probably would be more relevant to write about focusing on three OF the processes that has an influence, as other processes are discarded in this conceptual model and this is mentioned in the text.

Line 280: "*When fixing one location to the target site and varying the distance from the target site of the second location…*"
This sentence appears again somewhat confusing to me. Do you actually average over "three" locations here or only two? You refer to fixing the core to the target site (first core), and then refer to the "second site". What then denotes "distance of first core " in the figures (like Fig 6)?

Line 307: "*Our results which we obtained from analysing the climate model data and substantiated with our conceptual model provide guidance on where to drill N = 1,2,3 or more ice cores, or from which locations…*"

This statement is not entirely correct, the presented results tell about the relative distances (dimensions) of the core network optimal for the model, rather than point to specific locations that need to be derived via modelling for every target region.

Line 311: *"However, it is unclear whether these results can be one-to-one transferred to the real world, since they might depend on dynamical processes in the atmosphere which could differ between climate states or depend on initial conditions."* Consider adding "…or unaccounted model deficiencies"

Line 328: "*we expect the optimal spatial configuration to be more dependent on the study region*" … and very likely on the GCMiso model used in the analysis.

Line 331:" *We thus need to create an isotope record that*" Consider adding "As a proof of concept"

Line 352: …*we expect similar results to hold for other parts of Antarctica, and potentially also for other large-scale ice-coring regions such as Greenland"*
One can add that this is conditional on a simplified assumption of a nearly anisotropic exponential decorrelation scale length to be valid

Figure 3: Why the correlation value for a cell at approximately 70 S and 20E stands out?

Figure 6:  The caption is somewhat confusing. Is the "target site" also to be sampled or not? If this is the case, should this be a 3-dimensional case or not?

Figure 7: what is "Rank" on y-axis? Ranking according to the maximum correlation attained? It should than be mentioned explicitly.

 Figure B1, caption.
*"Note that the plots (a) and (c) are based on the same parameters and therefore identical".*
Why and where they are identical? This is not evident from the plots.

---

## Author Comment (AC1) · 4 Feb 2021

Author Reply to the Review Comments by **Lenneke Jong (Referee #1)**

on the manuscript

**How precipitation intermittency sets an optimal sampling distance for temperature reconstructions from Antarctic ice cores**

by Thomas Münch, Martin Werner, and Thomas Laepple,
submitted to *Climate of the Past* (https://doi.org/10.5194/cp-2020-128).

Thank you very much, Lenneke Jong, for the time you spent on reading and reviewing our manuscript. Below we include a point-by point response to both the general and all specific comments. The original referee comments are set in normal black font, our replies in blue, and suggested changes to the manuscript are shown by citing the manuscript text on a gray background with changes in red.
* * *
**General Comments:**

This manuscript investigates the possibility of selecting an optimal set of ice core locations to best reconstruct the temperature record at a target Antarctic site. They use an isotope enabled climate model to derive what they term a "sampling correlation structure" of sampling locations from concentric rings radiating out from a target site. Their findings that the optimal reconstruction can be obtained by combining a local core with another 500-1000km away to decrease the noise in the records due to precipitation intermittency is particularly interesting.

Thank you; this indeed might be the most striking result.

The paper itself is quite well structured, however, I think it would benefit from including the details of the conceptual model earlier into the main text of the paper when the decorrelation length scales relating to the sources of noise are introduced. A diagram would also be of great help. It is probably beyond the scope of this paper but I would have liked the authors to comment on the relative importance of the different sources of noise in the isotopic signal and which ones are dominant depending on location in Antarctica (eg coastal vs inland).

We do not fully agree. We think the reviewer refers to the introduction where we discuss the different noise sources (P2 LL27–45). However, firstly we argue here already that each noise source should exhibit a characteristic spatial scale of influence or decorrelation length. Secondly, while we think that, based on the overall review comments, the conceptual model does need some more introductory motivation, we do not think that the introduction is the right place for mentioning already more details of the model. We also think that the methods section is neither appropriate, since the conceptual model is used in order to interpret our results, and not as a method to produce the main results. We therefore suggest that we give more space to motivating the conceptual model and our assumption of radial symmetry at the beginning of section 4.1. Additionally, we will add the following figure to the appendix which graphically illustrates the different decorrelation lengths and processes in the conceptual model:

[Figure]

**Fig. 1.** Illustration of the decorrelation lengths in the conceptual model. Shown are as a function of distance the correlation between two temperature time series (black), between a temperature and a precipitation-weighted temperature time series (dashed black), between the intermittency noise (violet), between two precipitation-weighted temperature time series (green), and between a target temperature time series and the average of two precipitation-weighted temperature time series (orange), one located at the target site and one located a certain distance away from the target site as indicated on the x axis. Parameters are taken from the DML region.

Regarding the relative importance of the different noise sources: Previous studies have shown for the EDML site in East Antarctica that stratigraphic noise amounts to approximately up to 50 % of the total isotope variance at the seasonal time scale (Münch et al., 2016), however, quantitative estimates for other Antarctic regions are still missing. A similarly high relative contribution is expected from precipitation intermittency (Laepple et al., 2018), which probably has a larger impact further inland than compared to coastal regions (Casado et al., 2020). We will add a short discussion of these results to the introduction.

I found some of the discussions of study regions and target regions quite confusing and could do with some clearer explanations. I also thought a motivating example with actual ice core data at one of the target sites to demonstrate the reduction in noise would be very helpful, and would be useful in quantifying how much improvement in the temperature reconstruction is obtained with additional cores when all the other confounding factors are included in the isotope signal. They state that including these sources of noise, eg isotope diffusion is outside the scope of this study, but and I would like to see how the results hold up when the real data is included.

We are sorry that some of our explanations regarding study regions and target sites were not clear enough and we will improve the text as illustrated in our answers to the specific comments.

We agree that testing our results on real ice core data would be an ultimate goal. Such a test would include to find appropriate ice core data and suitable instrumental temperature records, which are sparse on Antarctica. We thus think that this is clearly a study on its own and much beyond the scope of this manuscript, also given the manuscript's current length and number of figures.

The authors are clear in the assumptions going in to the conceptual model and the analysis overall. They are also clear on the limitations of trying to use this optimal sampling correlation structure in the real world.

Overall, the manuscript is written clearly, and is of good quality and is of scientific interest, especially to the ice coring an palaeoclimate community. I have listed a few specific comments below that I believe would help improve the readability of the paper and recommend publication once these issues

are addressed.

Thank you. We are happy about this positive evaluation and are confident that addressing the specific comments will further improve the manuscript.

**Specific Comments:**

Lines 13-14: Remove final sentence as I didn't see the application of this technique to anything other than temperature reconstructions from ice cores discussed.

You are correct that in the paper our technique for assessing the optimal sampling strategy is only applied to temperature reconstructions from ice cores. Therefore, this final sentence of the abstract was meant to be an outlook to emphasise that our technique is however general enough to be extended to other palaeoclimate (temperature) proxies that face similar problems, i.e. noise sources working on different spatial scales. In addition, we do pick up this topic in the final part of the Conclusions (LL 353–357).

Unless the editor explicitly disagrees, we would keep this sentence/part in the manuscript as a reference for the broader palaeoclimate community, but we suggest to rephrase the mentioned sentence to more clearly articulate that it is being meant as an outlook for further studies:

[...] It also broadens our knowledge on the processes that shape the isotopic record and their typical correlation scales. Finally, many palaeoclimate reconstruction efforts face the similar challenge of spatially correlated noise, and our presented method could directly assist further studies in determining optimal sampling strategies also for these problems.

And in the Conclusions (L355 et seq.):

This likely applies to various marine as well as terrestrial proxy types, and our strategy thus might offer a step forward in the best use of sampling and measurement capacity for quantitative climate reconstructions, which needs to be investigated in further studies.

section 2.3.3: Some motivation and justification for the sampling scheme of rings and selection would be helpful. It was also not clear at times if N was referring to the number of grid cells, or rings and also is presumably different to $N_{grid}$ mentioned later on line 132 onwards.

We chose the sampling scheme of rings since it provides a computationally efficient way to estimate a statistically solid average correlation as a function of distance between the averaged locations; see also our more detailed answer on the similar issue raised by the second reviewer.

In general, in the manuscript the symbol $N$ (with or without subscript) always refers to a particular number of model grid cells, but we agree that our use of this terminology may have been unclear at times, since there are different contexts in which we use this symbol. To make it clearer for the reader, we suggest to adjust the terminology as follows:

- In section 2.1 (LL82–84), the number $N_{grid}$ = 442 refers to the total number of grid cells which lie, in our specific model simulation, on the Antarctic continent and which are used for all our analyses (since, obviously, you cannot drill ice cores on marine sites). Since this number is not referred to at any later stage in the manuscript, we suggest to drop the term $N_{grid}$ here in L84:
  [...] extracted from the total number of 442 model grid cells that are available for the Antarctic continent (Münch and Werner, 2020).
- In section 2.3.4 (LL129–139), the number $N_{grid}$ refers to the number of grid cells which lie within our defined DML and Vostok study regions and which are used as temperature target sites. To improve clarity, we will, similar to above, refer to the number of grid cells within the two study regions explicitly but without using the symbol $N_{grid}$, and will rephrase the respective text passages as follows:
  (L132) We define the DML region as the area of ±17.5° longitude and ±5° latitude around the European Project for Ice Coring in Antarctica (EPICA) DML site (EDML; 75° S, 0° E; Fig. 1), consisting of 26 model grid cells.
  (L135) For the Vostok region, we choose an identical latitudinal and longitudinal coverage ($N_{grid}$=

30) with respect to the Vostok station (78.47° S, 106.83° E; Fig. 1), covering 30 model grid cells and encompassing the sites of the deep Vostok and Dome C ice cores and of several shallower cores (Stenni et al., 2017), [...].

- Throughout the manuscript, the term $N$ (i.e. without subscript) is used to refer to the number of grid cells (i.e. time series at the locations of these grid cells) which are averaged and correlated with a target site temperature time series. To improve clarity, we will replace $N$ by $N_\ell$ ($\ell$ for "locations") and introduce this terminology explicitly in section 2.3.1.

section 3.1 I would have liked to see what level of significance is attached to each correlation coefficient reported.

The correlation values are all highly significant; please see our answer below to your remark on Fig. 3.

Fig 2: (And other figures) Use of diverging colour map for only positively increasing correlation or lengths scales is a bit distracting. Consider only using reds for instance.

We agree and will use a sequential color scheme (red hue) for Fig. 2 and for the correlation map figures (Figs. 3, 4, 6, A1, and A2).

Fig 3. This is an important figure in that it shows the precipitation weighting being important in the correlation. I would like to see some indication of where these correlations are significant to (eg p<0.05). It would be nice to see another map explicitly showing the difference in correlation coefficients between the two, as it looks to be some regions where there is no difference at all.

We will add the map showing the differences in correlation (Fig. 2 below) to the manuscript as part of manuscript Fig. 3. This map nicely illustrates that the correlations tend to remain unaffected mostly in the coastal regions of Antarctica, where precipitation intermittency is expected to be less important (Casado et al., 2020).

However, indicating the significance of the correlation coefficients in the maps does not make sense statistically, since the correlation values are all highly significant (p <<< 0.01) given the long time series (1200 data points). Even if one accounted for autocorrelation of the data, the significance should still remain very high across all grid cells.

[Figure]

**Figure 2.** The difference in correlation coefficient between using unweighted and precipitation-weighted 2 m temperature time series for the correlation with the local 2 m temperature (i.e. the difference between manuscript figure 3b and 3a).

3.3 Optimal ice-core sampling structures line 187 "we compute the mean of correlation results obtained between a target site temperature and individual grid cells in order to reduce local variability in the model data" Is this what you really mean, averaging the correlations? This seems like a strange thing to do if you are looking to maximise the correlation overall?

We agree that our explanation might be be ambiguous or misleading here. Of course, we first average the $N$ isotope time series, taken from $N$ grid cells, and then compute the correlation of this averaged record with the target site temperature time series. We iterate this approach over all possible combinations (or over a finite number of Monte Carlo combinations) of drawing $N$ grid cells from a given ring combination, and only then average the correlations from these iterations to obtain the mean correlation for this ring combination. We then analyse the next ring combination. This approach is in more detail explained in Sect. 2.3.4. We will revise the text in Sect. 3.3. to better and unambiguously summarise this approach. In addition, we will take care that the methods text in Sect. 2.3.4 is also clearly understandable.

I would also like more comment on how much of the regional difference is lost by averaging to only get a function of radial distance. As the DML and Vostok results differ and you suggest there are regional differences. Surely the correlations are not radially uniform?

We agree that the correlations do not need to be radially uniform. From physical arguments we expect, however, that the first-order spatial correlation patterns are largely invariant against rotation. This is indeed the case for the temperature field, as the correlation maps show for the EDML and Vostok sites (Fig. 3 below), and similar patterns are observed for other Antarctic regions. However, for $\delta^{18}O$, and also partly through the effect of precipitation weighting, indeed rather strong radial asymmetry can occur. Nevertheless, still a contribution from radially symmetric patterns may exist, and our approach of a radial averaging is based on assuming such symmetric contributions. This can be motivated by the fact that in real world applications one may not necessarily know in which direction the correlation pattern is maximal, so that radial symmetry is the most straightforward assumption. And indeed our results suggest that a gain in correlation can be achieved nevertheless, when we use optimal core locations based on an analysis assuming radial symmetry, despite the actual form of the spatial correlation patterns.

a.

[Figure]

b.

[Figure]

**Figure 3.** Smoothed maps of the correlation between the temperature time series at the target sites EDML (a) and Vostok (b) and different climate model fields: temperature, precipitation-weighted temperature, $\delta^{18}$O and precipitation-weighted $\delta^{18}$O (from upper left to lower right panel, respectively). Note that the correlation patterns for temperature (upper left panels in a and b) are actually nearly radially symmetric around the target site and only appear elongated in North–South direction due to the map projection. For inclusion of this figure into the manuscript we would use a proper colour scheme and a polar projection, similar to manuscript Fig. 4.

At this point, we think it would be most beneficial for the reader to include the presentation of Fig. 3 in the manuscript (e.g. in a new subsection after 3.1) and to use it as a starting point to motivate the ring sampling in section 3.3 as well as for the discussion (Sect. 4.1). However, this would of course extend the manuscript substantially and it would also increase the number of figures, and we thus would leave it to the editor to decide whether this extension is reasonable.

Fig 4. It would be also useful to have the concentric circles marked around the target sites. Can you also comment on why the locations shift when more cores locations are added? That is, is the location in the N=1 case also included in the N=3 case as it looks as though they have moved slightly. These segments don't seem to correspond to the regions mapped out by the black polygons in Fig. 1 so it is not clear to me what the study regions actually are. Does it mean that the cores in the N>1 cases can be from outside of those black polygons too as appears to be the case here?

We use the study regions (the black polygons in Fig. 1) only to define regions from which we select temperature target sites, with respect to which we conduct our analyses and across which the results are then averaged to obtain regional estimates, such as for Figs. 5, 6, and A1. Here, for Fig. 4, we use only a single temperature target site from within the study regions, namely the EDML site (Fig. 4a–c) and the Vostok site (Fig. 4d–f). The $\delta^{18}$O$^{(pw)}$ model grid cells that we average and correlate with the target temperature time series can, however, indeed be selected from a wider region than the study region, namely from the 2000-km circles around the target sites (as explained in Sect. 2.3.2).

We will add the line of the 2000-km circles around the target site to the plots and explain in the figure caption that the $\delta^{18}$O$^{(pw)}$ grid cells ("ice cores") can be chosen from within these circles.

Yes, you correctly observe that the optimal location (model grid cell) in the N=1 case for EDML is no longer included in the N=3 or N=5 case, while this is the case for Vostok. However, this is simply by chance: while for N=1 it is computationally easy and fast to find the best correlating grid cell within the 2000-km circles and so panels (a) and (d) display the "true" optimal location, for N=3 and N=5 we need to randomly select and average grid cells, using $10^5$ iterations. The displayed locations is the best configuration of these iterations, but does not necessarily need to be the "true" optimal configuration. However, in terms of correlation value with the target site temperature, the value from the best iteration

should be very close to the correlation value for the true optimal configuration due to the large number of performed iterations.

Fig 5. The black dashed line indicating the exponential fit is not included in the figure legend, only the caption. The dots in the plot don't appear to be at the expected 0, 250, 500km marks, but are a bit offset, is this deliberate? Again, I question the averaging step around the whole 250km rings, but it would be nice to see the spread in the correlation as a function of distance too. I am confused at what is meant by "all respective target sites in the DML and Vostok region", how are there more than one target sites for each region, are these different that the two crosses shown in Fig. 4?

We will add the explanation for the dashed line to the figure legend and include the correlation scatter (standard deviation of the correlation results across the different target sites within the study regions); see the revised plot in Fig. 4 below. Yes, the dots in the plots are deliberately placed in the middle of the ring bin borders, i.e. at distances of 125, 375, 625, ... km, since each correlation value is an average value across a ring bin (from 0–250, 250–500, 500–750, ... km) by averaging the individual correlations obtained between the target site in the centre and all grid cells that lie within each bin.

Regarding the target sites your comment shows that there is a clear need for us to better explain this concept in the revised manuscript. We use "target site" as the term to denote a model grid cell from which we use the temperature time series ($T_{2m}$) to correlate all other grid cells and variables with, and which defines the centre of the ring bins. For a single target site we can study the average correlation with distance similar as shown in Fig. 5. But Fig. 5 involves a second averaging step: To improve statistics, and to obtain regional estimates, we use all model grid cells within our defined DML and Vostok regions as target sites, one after the other, to get the correlation–distance dependencies for each target site and for each variable ($T_{2m}$, $\delta^{18}O$, $\delta^{18}O^{(pw)}$). Then we average all these 26 (DML region) and 30 (Vostok region) curves that we obtain for each variable, respectively, to produce the curves shown in Fig. 5. The same approach is used for the results shown in Figs. 6 and A1.

We will expand the explanations in sections 2.3.1 and 2.3.4 to clarify the concept of "target sites" and of "the averaging across target sites" to the reader.

[Figure]

**Figure 4.** Revised version of manuscript Fig. 5 with an estimated regional scatter of the correlation coefficient included.

Fig 6. These are interesting and show the clear difference for the precipitation weighted $\delta^{18}O$ and $T_{2m}$ for DML, but why is the Vostok case relegated to the appendix? The fact that they are different is an interesting result.

We fully agree that the difference between the DML and Vostok regions is an interesting result. When writing up the manuscript, we were concerned that we could overload the results section with too many coloured contour plots, which is why we moved the Vostok plots to the appendix. However, we are happy to combine both regions into a single Figure 6 (similar to Fig. B1) and remove Appendix A.

Fig 7. The way this figure is arranged, is the top row, marked rank 5, that which has the max correlation, or is the rank 1 row the highest correlation row? I find it very curious that in the Vostok region the optimal arrangement comes with no local sites in many of the cases

The maximum correlation correponds to the combination/row that is marked as rank 1; we will add a clarifying remark to the figure caption.

Yes, it is indeed a curious result that in the Vostok region the optimal arrangement comes mostly without local sites, but we see no evidence for not trusting this result in view of the agreement between N = 3 and N = 5 and given the statistics from the large number of sampled locations in our ring sampling scheme.

line 220: Suggests that regional differences play a part here and I would like a comment on what those differences are (elevation, distance from coast etc?)

We can only speculate what possible reasons could there be for the difference in temperature–isotope correlation between the DML and Vostok regions (see Fig. 5 in the manuscript). One factor might indeed be the larger elevation and distance from the coast, i.e. a stronger continentality at Vostok, and related to this, differences in the distillation paths of the transported vapour.

Line 232: Does the averaging have an effect on the significance of the correlations?

As pointed out above, there is statistically no sense in studying the significance of the correlation coefficients given that each time series has such a large number of data points.

Fig. 8 (b) I assume the colours on the histogram are the same as in (a), but please add the legend anyway. Can you comment on the low correlation outliers for the EDML case.

Yes, the colours are the same in (b) as in panel (a); nevertheless, we will add a second legend to panel (b). The low correlation outliers in the EDML case stem from the grid cell combinations which include one anomalous grid cell located at ~ 72.4 °S, 22.5 °E; see Fig. 3 and our reply to a respective comment by reviewer #2. We will have a deeper look into this issue.

Section 4.1 I found most of the discussion very clear and thorough, but re-iterate that a good schematic diagram to illustrate the length scales in the conceptual model would be very useful.

Thank you; please see our reply to your General Comments and Fig. 1 there.

References:

Casado, M., Münch, T., and Laepple, T.: Climatic information archived in ice cores: impact of intermittency and diffusion on the recorded isotopic signal in Antarctica, Clim. Past, 16, 1581–1598, https://doi.org/10.5194/cp-16-1581-2020, 2020.

Laepple, T., Münch, T., Casado, M., Hoerhold, M., Landais, A., and Kipfstuhl, S.: On the similarity and apparent cycles of isotopic variations in East Antarctic snow pits, The Cryosphere, 12, 169–187, https://doi.org/10.5194/tc-12-169-2018, 2018.

Münch, T., Kipfstuhl, S., Freitag, J., Meyer, H., and Laepple, T.: Regional climate signal vs. local noise: a two-dimensional view of water isotopes in Antarctic firn at Kohnen Station, Dronning Maud Land, Clim. Past, 12, 1565-1581, https://doi.org/10.5194/cp-12-1565-2016, 2016.

---

## Author Comment (AC2) · 4 Feb 2021

Author Reply to the Review Comments by **Dmitry Divine (Referee #2)**

on the manuscript

**How precipitation intermittency sets an optimal sampling distance for temperature reconstructions from Antarctic ice cores**

by Thomas Münch, Martin Werner, and Thomas Laepple,
submitted to *Climate of the Past* (https://doi.org/10.5194/cp-2020-128).

Thank you very much, Dmitry Divine, for the time you spent on reading and reviewing our manuscript. Below we include a point-by point response to both the major and all minor comments. The original referee comments are set in normal black font, our replies in blue, and suggested changes to the manuscript are shown by citing the manuscript text on a gray background with changes in red.
* * *
**Major Comments:**

My major comment concerns the presentation of the sampling procedure in 2.3.2-2.3.3, which I have found not very straightforward to comprehend. One should admit I have spent quite some time trying to understand the actual details behind the technique, though this difficulty could of course be quite individual. The grip of understanding came later, while reading "Results", yet some questions still remain. A number of minor questions that emerged while reading the manuscript, could therefore be a result of my unclear understanding of the basics of the proposed method.

We are sorry and apologise for the fact that our description of the sampling procedure was not straightforward to comprehend and are grateful to the reviewer that he still spent the time trying to understand our approach. In addition to implementing our suggestions in the below answers to the specific comments, we will thoroughly go through the overall methods text again and revise it in order to improve formulation and clarity. Furthermore, we suggest to include the following figure as an additional part of manuscript Fig. 1 to visualise our approach and hope that is helpful for aiding comprehension:

[Figure]

**Fig. 1** From an array of grid cells (grey points), we choose sets of grid cells, consisting of $N_\ell$ cells, around a target site (black cross), which are drawn from radial bins determined by selected combinations of rings (red). As an example for $N_\ell = 2$, possible grid cell sets are shown for the cases of (i) combining the innermost ring with itself (grid cells marked black), (ii) combining the innermost ring with the second ring (grid cells marked blue), and (iii) combining the third and the fourth ring (grid cells marked orange).

I would like to note also that sometimes the discussion around/use of terms like "target site" or "local site" may appear confusing, same as the actual dimension of the core network being discussed. May be some simplification/clarification of 2.3. can improve the readability?

The target site is always the site from which we use the pure temperature time series as a reference or, in other words, which is sought to be reconstructed from the "ice core" network. We will clarify this in the methods sections. "Local site" is loosely referred to as a site close to the target site, so e.g. the target site grid cell itself or grid cells within the innermost ring. We will go through the manuscript text to check whether certain instances of this use can be improved in terminology. Finally, the "study regions" are areas from which we sequentially use all contained grid cells as target sites. The results with respect to each target site in a study region are then averaged in order to arrive at regional estimates, such as for Figs. 5, 6, and A1. We will clarify this in Section 2.3.4.

In general, with respect to the sampling strategy, the question is why the authors initiated the procedure with these concentric rings used for spatial sampling, instead of just random seeding of the "sampling locations", calculating the metrics of interest and then ordering them according to the distances between the locations? It sounds way more straightforward to comprehend than via introducing these circular sampling areas with an increment of an arbitrary choice.

Yes, we agree that the approach of randomly seeding the sampling locations and subsequent ordering of the results according to the distances between the locations is, conceptually, a more straightforward procedure. In fact, our approach for N = 1, i.e. sampling one location only, is identical to the random seeding approach, if the latter approach uses a sufficient number of iterations to sample the entire required space.

However, this is also the critical point why we chose the different approach of the ring sampling scheme. While for N = 1 it is compuationally easy and fast to sufficiently sample the required space (e.g. a 2000 km circle around a target site) by random seeding, this is more problematic for N >= 2. Here, the total number of possibilities of combining two, or more, grid cells is much larger than the actual number of grid cells (the more the larger N is), and a random seeding approach of these many possibilities will be strongly limited by available computation time. This will likely lead to an undersampling of the possible relative distances, especially for distances farther away from the target site due to the radially increasing number of grid cells.

Our ring sampling approach circumvents this undersampling: i) Since we sample all possible ring bin combinations, we ensure to sample the entire available sampling space of (binned) relative distances. ii) For each ring combination, we either sample all possible grid cell combinations (N = 1 and N = 2), or we sample a fixed (but large) number of randomly chosen grid cell combinations (N > 2). This ensures that either the entire availble sampling space is actually sampled, or – at least – a fixed number of grid cell combinations for every ring bin combination, so that the expectation value for every ring bin combination builds on the same number of combined grid cells.

We will add this motivation to the manuscript in section 2.3.3.

Also, as a suggestion for the future work, it would be highly useful to test the concept of this method on a different model with enabled stable water isotopes in precipitation in order to see how different the results/inference can be. Testing on the existing ice core network can be fairly problematic due to all the deficiencies (both in the available ice core and instrumental data) mentioned throughout the text.

This is a very good point. Indeed, we tested our concept also on the ECHAM6/MPI-OM-wiso PI-control simulation (unpublished) and obtained comparable results. However, this might also be due to the fact that both models use the same isotope scheme. So testing our results with a completely different isotope-enabled climate model is needed. We will add this suggestion to the conclusions of our manuscript.

**Minor Comments (the manuscript text shown in italics):**

Line 112: "*...define consecutive rings around this site with a 250 km radial width...*"
Here you refer to these concentric rings with a radius increment of 250 km, used for delimitation of the sampling regions, do I get it right? May be it needs to be specified already here. Can you also provide

any rationale behind the value of 250 km?

Yes, we here refer to the concentric rings (red circles in Fig. 1) with 250 km radial increments which we use to sample grid cell combinations as a function of whether they lie within the same ring or within different rings. We will refine the text here to make this clearer to the reader. We chose the value of 250 km radial extent as a trade-off between achieving a high spatial resolution and ensuring that a sufficient number of grid cells actually lie within the ring bin borders; e.g., the first ring (0–250 km) includes with respect to the EDML site already five grid cells only. Using a smaller radial extent (higher spatial resolution) would thus not be meaningful and would result in statistically less robust results.

Line 118: "*Finally, we report the mean correlation for every ring combination by averaging across all correlations of the analysed grid-cell combinations.*" Is this averaging based on the distance between the locations, or just everything? How then the distance-based value is calculated?

The averaging is performed across all analysed grid cell combinations for a given ring combination. The distance information is then "only" given by the radial midpoint distances of the combined ring bins relative to the target site. We will add a sentence here to clarify how we obtain this distance information.

Line 119: "*...for sampling N locations from the model field depending on the distances between the locations.*" See my previous comment. If everything is averaged out, how the distance based sorting/ranking is implemented?

We will clarify here how we obtain the distance information; see our previous answer.

Line 136: *"(−78.47 S)»*. No need in "-" before the latitude value if "S" is explicitly indicated.

Thanks for spotting this inconsistency; we will correct this.

Line 184: "*...depend on the specific simulated climate state or result...*"
It would be meaningful to add that it also includes the actual model used and the stable water isotope scheme applied in the model.

We will add the possible dependence on the climate model isotope scheme here.

Line 212: "*...maximum average correlation is to sample one location from the innermost ring and the second location from the fifth ring*"
this is not entirely apparent as both maxima in Fig 6 seem to be found on the "5th ring".

We are afraid that this is a misunderstanding. There are indeed two maxima visible in Fig. 6c, both between 1000 and 1250 km (i.e. fifth ring), one along the x axis and one along the y axis. However, this is the same information since the locations of the two cores are indistinguishable, i.e., it doesn't matter whether we put the "first core" within the fifth ring and the "second core" within the first ring (maximum along the x axis) or vice versa. In other words, the figure is symmetric along the diagonal and half of the plot already contains the full information. We chose this way of presentation for aesthetic reasons. We will add a sentence to the caption noting that the plot is symmetrical since the locations of the cores are indistinguishable.

Line 255: "*For a conceptual model of the sampling correlation structure, we focus on three processes that influence...*". It is probably would be more relevant to write about focusing on three OF the processes that has an influence, as other processes are discarded in this conceptual model and this is mentioned in the text.

We will edit the text as suggested.

Line 280: "*When fixing one location to the target site and varying the distance from the target site of the second location...*"
This sentence appears again somewhat confusing to me. Do you actually average over "three" locations here or only two? You refer to fixing the core to the target site (first core), and then refer to the "second

site". What then denotes "distance of first core " in the figures (like Fig 6)?

We are sorry for this ambiguity. We indeed average across only two locations. What is meant here in general is to fix one core (the "first core") to the innermost location and only vary the location of the second core. In the conceptual model, which is discussed at this point, the innermost location is identical to the centre of the rings, i.e. the target site (simply due to the fact how the conceptual model is set up numerically). In the climate model results (Figs. 6, A1), however, the innermost location in the ring sampling scheme can only be obtained for putting the first core within the first (innermost) ring (0–250 km). We will revise the text accordingly here:

"When fixing the position of one core to the innermost location..."

and at other respective passages to avoid this ambiguity.

Line 307: "*Our results which we obtained from analysing the climate model data and substantiated with our conceptual model provide guidance on where to drill N = 1,2,3 or more ice cores, or from which locations...*"
This statement is not entirely correct, the presented results tell about the relative distances (dimensions) of the core network optimal for the model, rather than point to specific locations that need to be derived via modelling for every target region.

We do not fully agree. If one believes in the "picking" results of directly analysing the best grid cell combinations (Section 3.2 and Fig. 4), our results can directly advise where to drill cores for a specific target site. This is also elaborated in the following paragraph (LL310–315). But you are correct that one would need to do the analysis for every specific target site one is interested in. In this regard, the ring sampling results are more general since they should apply to a larger region (DML, Vostok region) but with the downside that the results only tell us about the optimal relative distances of the core network, as you correctly observe. We will revise the introductory paragraph of this section in order to better reflect these two different views.

Line 311: "*However, it is unclear whether these results can be one-to-one transferred to the real world, since they might depend on dynamical processes in the atmosphere which could differ between climate states or depend on initial conditions.*" Consider adding "...or unaccounted model deficiencies"

We will add this additional information.

Line 328: "*we expect the optimal spatial configuration to be more dependent on the study region*" ... and very likely on the GCMiso model used in the analysis.

Here, we talk specifically about the results for three or more ice cores (N >= 3); your statement rather applies to the results in general. We think it is thus sufficient to revise the text around L184, as suggested above in the respective comment.

Line 331:" *We thus need to create an isotope record that*" Consider adding "As a proof of concept"

We will add the suggested phrase.

Line 352: ...*we expect similar results to hold for other parts of Antarctica, and potentially also for other large-scale ice-coring regions such as Greenland*"
One can add that this is conditional on a simplified assumption of a nearly anisotropic exponential decorrelation scale length to be valid

We will add this limitation.

Figure 3: Why the correlation value for a cell at approximately 70 S and 20E stands out?

Inspection of the time series of this grid cell located at ~ 72.4 °S, 22.5 °E shows that the isotope time series exhibits one anomalous time step where the delta value erroneously rises far above 0 permil. This causes the outlier correlation with the temperature time series. We will remove this anomalous

time step from the isotope time series to fix this, and we will carefully check whether there are further anomalies.

Figure 6: The caption is somewhat confusing. Is the "target site" also to be sampled or not? If this is the case, should this be a 3-dimensional case or not?

No, it is the 2-dimensional case, i.e. averaging two locations, as explicitly stated in the first and second sentence of the caption. The grid cell of the target site lies in the centre of the innermost ring, so it is (implicitly) included in the analyses when sampling is performed for combining the innermost ring with itself (distance of both cores <250 km) or with one of the other rings (distance of only one of the cores <250 km). But please note that the results are the average across all grid cell combinations for a specific ring combination, hence the target site grid cell only contributes proportionately to the overall average value for the combinations which include the innermost ring.

Figure 7: what is "Rank" on y-axis? Ranking according to the maximum correlation attained? It should than be mentioned explicitly.

Yes, the "Rank" means ranking according to the maximum attained correlation, with Rank 1 denoting the case with the highest correlation. We will add this information to the figure caption.

Figure B1, caption.
*"Note that the plots (a) and (c) are based on the same parameters and therefore identical".*
Why and where they are identical? This is not evident from the plots.

Thanks for spotting this typo, correct is that plots (a) and (d) are identical. We will fix this.

---

## Author Comment (AC4) · 9 Feb 2021

Dear Lenneke, dear Nerilie,

I have just realised there is a typo in the caption for Figure 2 in our reply to the reviewer comment; the correct caption should read:

"The difference in correlation coefficient between using unweighted and precipitation-weighted 2 m temperature time series for the correlation with the local oxygen isotope composition (i.e. the difference between manuscript figure 3b and 3a)."

I apologise for the confusion this might have caused.

Kind regards, Thomas

---

## Author Response (AR1)

on the manuscript

**How precipitation intermittency sets an optimal sampling distance for temperature reconstructions from Antarctic ice cores**

by Thomas Münch, Martin Werner, and Thomas Laepple,
submitted to *Climate of the Past* (https://doi.org/10.5194/cp-2020-128).
* * *
Dear Nerilie,

please find attached below our point-by-point responses to the two reviews and the one short comment on our above-mentioned manuscript, which describe in detail the changes we implemented to address the issues raised by the reviewers. We are confident that these changes constitute a significant improvement of the paper and are looking forward to your decision.

With best wishes,
on behalf of my co-authors,

Thomas Münch

Author Reply to the Review Comments by **Lenneke Jong (Referee #1)**

on the manuscript

**How precipitation intermittency sets an optimal sampling distance for temperature reconstructions from Antarctic ice cores**

by Thomas Münch, Martin Werner, and Thomas Laepple,
submitted to *Climate of the Past* (https://doi.org/10.5194/cp-2020-128).

Thank you very much, Lenneke Jong, for the time you spent on reading and reviewing our manuscript. Below we include a point-by point response to both the general and all specific comments. The original referee comments are set in normal black font, our replies in blue, and the changes made to the manuscript are shown by citing the manuscript text on a gray background with revised text highlighted in red.
* * *
**General Comments:**

This manuscript investigates the possibility of selecting an optimal set of ice core locations to best reconstruct the temperature record at a target Antarctic site. They use an isotope enabled climate model to derive what they term a "sampling correlation structure" of sampling locations from concentric rings radiating out from a target site. Their findings that the optimal reconstruction can be obtained by combining a local core with another 500-1000km away to decrease the noise in the records due to precipitation intermittency is particularly interesting.

Thank you; this indeed might be the most striking result.

The paper itself is quite well structured, however, I think it would benefit from including the details of the conceptual model earlier into the main text of the paper when the decorrelation length scales relating to the sources of noise are introduced. A diagram would also be of great help. It is probably beyond the scope of this paper but I would have liked the authors to comment on the relative importance of the different sources of noise in the isotopic signal and which ones are dominant depending on location in Antarctica (eg coastal vs inland).

We do not fully agree. We think the reviewer refers to the introduction where we discuss the different noise sources (P2 LL27–45). However, firstly we argue here already that each noise source should exhibit a characteristic spatial scale of influence or decorrelation length. Secondly, while we think that, based on the overall review comments, the conceptual model does need some more introductory motivation, we do not think that the introduction is the right place for mentioning already more details of the model. We also think that the methods section is neither appropriate, since the conceptual model is used in order to interpret our results, and not as a method to produce the main results. Instead, we revised the beginning of section 4.1 to give more space to motivating the conceptual model and our assumption of radial symmetry. Additionally, we added Fig. 1 below to the appendix (as a new Fig. A1), which graphically illustrates the different decorrelation lengths and processes in the conceptual model.

Regarding the relative importance of the different noise sources: Previous studies have shown for the EDML site in East Antarctica that stratigraphic noise amounts to approximately up to 50 % of the total isotope variance at the seasonal time scale (Münch et al., 2016), however, quantitative estimates for other Antarctic regions are still missing. A similarly high relative contribution is expected from precipitation intermittency (Laepple et al., 2018), which probably has a larger impact further inland than compared to coastal regions (Casado et al., 2020). We added a short discussion of these results to the introduction.

[Figure]

**Fig. 1.** Illustration of the decorrelation lengths in the conceptual model. Shown are as a function of distance the correlation between two temperature time series (black), between a temperature and a precipitation-weighted temperature time series (dashed black), between the intermittency noise (violet), between two precipitation-weighted temperature time series (green), and between a target temperature time series and the average of two precipitation-weighted temperature time series (orange), one located at the target site and one located a certain distance away from the target site as indicated on the x axis. Parameters are taken from the DML region.

I found some of the discussions of study regions and target regions quite confusing and could do with some clearer explanations. I also thought a motivating example with actual ice core data at one of the target sites to demonstrate the reduction in noise would be very helpful, and would be useful in quantifying how much improvement in the temperature reconstruction is obtained with additional cores when all the other confounding factors are included in the isotope signal. They state that including these sources of noise, eg isotope diffusion is outside the scope of this study, but and I would like to see how the results hold up when the real data is included.

We are sorry that some of our explanations regarding study regions and target sites were not clear enough and we improved the text as illustrated in our answers to the specific comments.

We agree that testing our results on real ice core data would be an ultimate goal. Such a test would include to find appropriate ice core data and suitable instrumental temperature records, which are sparse on Antarctica. We thus think that this is clearly a study on its own and much beyond the scope of this manuscript, also given the manuscript's current length and number of figures.

The authors are clear in the assumptions going in to the conceptual model and the analysis overall. They are also clear on the limitations of trying to use this optimal sampling correlation structure in the real world.

Overall, the manuscript is written clearly, and is of good quality and is of scientific interest, especially to the ice coring an palaeoclimate community. I have listed a few specific comments below that I believe would help improve the readability of the paper and recommend publication once these issues are addressed.

Thank you. We are happy about this positive evaluation and are confident that the implemented changes from addressing your specific comments further improved the manuscript.

**Specific Comments:**

Lines 13-14: Remove final sentence as I didn't see the application of this technique to anything other

than temperature reconstructions from ice cores discussed.

You are correct that in the paper our technique for assessing the optimal sampling strategy is only applied to temperature reconstructions from ice cores. Therefore, this final sentence of the abstract was meant to be an outlook to emphasise that our technique is however general enough to be extended to other palaeoclimate (temperature) proxies that face similar problems, i.e. noise sources working on different spatial scales. In addition, we do pick up this topic in the final part of the Conclusions (LL 353–357 in the discussion paper).

Therefore, we decided to keep the mentioned sentence of the abstract in the manuscript as a reference for the broader palaeoclimate community, but we rephrased it to more clearly articulate that it is being meant as an outlook for further studies:

[...] It also broadens our knowledge on the processes that shape the isotopic record and their typical correlation scales. Finally, many palaeoclimate reconstruction efforts face the similar challenge of spatially correlated noise, and our presented method could directly assist further studies in also determining optimal sampling strategies for these problems.

And we changed the last sentence of the Conclusions to:

This likely applies to various marine as well as terrestrial proxy types, and our strategy thus might offer a step forward in the best use of sampling and measurement capacity for quantitative climate reconstructions, which needs to be investigated in further studies.

section 2.3.3: Some motivation and justification for the sampling scheme of rings and selection would be helpful. It was also not clear at times if N was referring to the number of grid cells, or rings and also is presumably different to $N_{grid}$ mentioned later on line 132 onwards.

We chose the sampling scheme of rings since it provides a computationally efficient way to estimate a statistically solid average correlation as a function of distance between the averaged locations; see also our more detailed answer on the similar issue raised by the second reviewer.

In general, in the manuscript the symbol $N$ (with or without subscript) always refers to a particular number of model grid cells, but we agree that our use of this terminology may have been unclear at times, since there are different contexts in which we use this symbol. To make it clearer for the reader, we adjusted the terminology as follows:

- In section 2.1 (LL82–84), the number $N_{grid} = 442$ refers to the total number of grid cells which lie, in our specific model simulation, on the Antarctic continent and which are used for all our analyses (since, obviously, you cannot drill ice cores on marine sites). Since this number is not referred to at any later stage in the manuscript, we dropped the term $N_{grid}$ here in L84:
  [...] extracted from the total number of 442 model grid cells that are available for the Antarctic continent (Münch and Werner, 2020).
- In section 2.3.4 (LL129–139 of the discussion paper), the number $N_{grid}$ refers to the number of grid cells which lie within our defined DML and Vostok study regions and which are used as temperature target sites. To improve clarity, similar to above we now refer to the number of grid cells within the two study regions explicitly but without using the symbol $N_{grid}$, and rephrased the respective text passages as follows:

  We define the DML region as the area of ±17.5˚ longitude and ±5˚ latitude around the European Project for Ice Coring in Antarctica (EPICA) DML site (EDML; 75˚ S, 0˚ E; Fig. 1), consisting of 26 model grid cells.

  For the Vostok region, we choose an identical latitudinal and longitudinal coverage (N̶g̶r̶i̶d̶ ̶=̶ ̶3̶0̶) with respect to the Vostok station (78.47˚ S, 106.83˚ E; Fig. 1), covering 30 model grid cells, and encompassing the sites of the deep Vostok and Dome C ice cores, of several shallower cores (Stenni et al., 2017), [...].
- Throughout the discussion manuscript, the term $N$ (i.e. without subscript) has been used to refer to the number of grid cells (i.e. time series at the locations of these grid cells) which are averaged and

correlated with a target site temperature time series. To improve clarity, we replaced $N$ by $N_\ell$ ($\ell$ for "locations") and introduced this terminology explicitly in section 2.3.1, which we revised as a whole.

section 3.1 I would have liked to see what level of significance is attached to each correlation coefficient reported.

The correlation values are all highly significant; please see our answer below to your remark on Fig. 3.

Fig 2: (And other figures) Use of diverging colour map for only positively increasing correlation or lengths scales is a bit distracting. Consider only using reds for instance.

We agree and now use a sequential color scheme (red hue) for Fig. 2 and for all other respective figures.

Fig 3. This is an important figure in that it shows the precipitation weighting being important in the correlation. I would like to see some indication of where these correlations are significant to (eg p<0.05). It would be nice to see another map explicitly showing the difference in correlation coefficients between the two, as it looks to be some regions where there is no difference at all.

We added the map showing the differences in correlation (Fig. 2 below) to the manuscript, merging it with the discussion manuscript Figs. 2+3 to the new Fig. 2. The difference map nicely illustrates that the correlations tend to remain unaffected mostly in the coastal regions of Antarctica, where precipitation intermittency is expected to be less important (Casado et al., 2020).

However, indicating the significance of the correlation coefficients in the maps does not make sense statistically, since the correlation values are all highly significant ($p \lll 0.01$) given the long time series (1200 data points). Even if one accounted for autocorrelation of the data, the significance should still remain very high across all grid cells.

[Figure]

**Figure 2.** The difference in correlation coefficient between using unweighted and precipitation-weighted 2 m temperature time series for the correlation with the local precipitation-weighted $\delta^{18}O$ (i.e. the difference between manuscript figure 3b and 3a).

3.3 Optimal ice-core sampling structures line 187 "we compute the mean of correlation results obtained between a target site temperature and individual grid cells in order to reduce local variability in the model data" Is this what you really mean, averaging the correlations? This seems like a strange thing to do if you are looking to maximise the correlation overall?

We agree that our explanation might be ambiguous or misleading here. Of course, we first average the $N$ isotope time series, taken from $N$ grid cells, and then compute the correlation of this averaged record with the target site temperature time series. We iterate this approach over all possible combinations (or

over a finite number of Monte Carlo combinations) of drawing $N$ grid cells from a given ring combination, and only then average the correlations from these iterations to obtain the mean correlation for this ring combination. We then analyse the next ring combination. This approach is in more detail explained in the revised Sect. 2.3.2 (formerly 2.3.3). We revised the introductory text of Sect. 3.4 (formerly 3.3) to better and unambiguously summarise this approach.

I would also like more comment on how much of the regional difference is lost by averaging to only get a function of radial distance. As the DML and Vostok results differ and you suggest there are regional differences. Surely the correlations are not radially uniform?

We agree that the correlations do not need to be radially uniform. From physical arguments we expect, however, that the first-order spatial correlation patterns are largely invariant against rotation. This is indeed the case for the temperature field, as the correlation maps show for the EDML and Vostok sites (Fig. 3 below), and similar patterns are observed for other Antarctic regions. However, for $\delta^{18}O$, and also partly through the effect of precipitation weighting, indeed rather strong radial asymmetry can occur. Nevertheless, still a contribution from radially symmetric patterns may exist, and our approach of a radial averaging is based on assuming such symmetric contributions. This can be motivated by the fact that in real world applications one may not necessarily know in which direction the correlation pattern is maximal, so that radial symmetry is the most straightforward assumption. And indeed our results suggest that a gain in correlation can be achieved nevertheless, when we use optimal core locations based on an analysis assuming radial symmetry, despite the actual form of the spatial correlation patterns.

As we think that this discussion is most beneficial for the reader, we included the presentation of Fig. 3 in the manuscript as a new manuscript Fig. 3 and present it in a new subsection after Sect. 3.1. We also use it to motivate and interpret the ring sampling approach as the start for the discussion (now Sect. 4.1).

[Figure]

**Figure 3.** Spatial correlation to the temperature at the EDML and Vostok target sites. Shown are the correlations of the T2m time series at the target sites EDML (**a–d**) and Vostok (**e–h**}) with, respectively, the spatial fields of temperature (**a, d**), precipitation-weighted temperature (**b, f**), oxygen isotope composition (**c, g**), and precipitation-weighted oxygen isotope composition (**d, h**). The target sites are marked with a black cross, black lines indicate correlation contour lines incremented by 0.1.

Fig 4. It would be also useful to have the concentric circles marked around the target sites. Can you also comment on why the locations shift when more cores locations are added? That is, is the location in the N=1 case also included in the N=3 case as it looks as though they have moved slightly. These segments don't seem to correspond to the regions mapped out by the black polygons in Fig. 1 so it is not clear to me what the study regions actually are. Does it mean that the cores in the N>1 cases can be from outside of those black polygons too as appears to be the case here?

We use the study regions (the black polygons in Fig. 1) only to define regions from which we select

temperature target sites, with respect to which we conduct our analyses and across which the results are then averaged to obtain regional estimates, such as for Figs. 5, 6, and A1. We clarified this in Sect. 2.3.3 (formerly 2.3.4). Here, for Fig. 4, we use only a single temperature target site from within the study regions, namely the EDML site (Fig. 4a–c) and the Vostok site (Fig. 4d–f). The $\delta^{18}O^{(pw)}$ model grid cells that we average and correlate with the target temperature time series can, however, indeed be selected from a wider region than the study region, namely from the 2000-km circles around the target sites (as explained in Sect. 2.3.2). We added the line of the 2000-km circles around the target site to the plots and now explain in the figure caption that the $\delta^{18}O^{(pw)}$ grid cells ("ice cores") can be chosen from within these selection circles.

Yes, you correctly observe that the optimal location (model grid cell) in the N=1 case for EDML is no longer included in the N=3 or N=5 case, while this has been the case for Vostok (no longer, however, for the updated results). However, this is simply by chance: while for N=1 it is computationally easy and fast to find the best correlating grid cell within the 2000-km selection circles and so panels (a) and (d) display the "true" optimal location, for N=3 and N=5 we needed to randomly select and average grid cells, using $10^5$ iterations. The displayed locations is the best configuration of these iterations, but does not necessarily need to be the "true" optimal configuration. However, in terms of correlation value with the target site temperature, the value from the best iteration should be very close to the correlation value for the true optimal configuration due to the large number of performed iterations.

We note here that we were able to improve our "picking" code after our initial reply to this review so that the results are now exact up to N=3, only for N=5 our results are still based on randomly picking the grid cells (see the revised Sect. 2.3.2). The results displayed in the new Fig. 4 therefore slightly differ from the version of the discussion paper (in addition to the effect from the outlier filtering we now perform on the model data; see below).

Fig 5. The black dashed line indicating the exponential fit is not included in the figure legend, only the caption. The dots in the plot don't appear to be at the expected 0, 250, 500km marks, but are a bit offset, is this deliberate? Again, I question the averaging step around the whole 250km rings, but it would be nice to see the spread in the correlation as a function of distance too. I am confused at what is meant by "all respective target sites in the DML and Vostok region", how are there more than one target sites for each region, are these different that the two crosses shown in Fig. 4?

We added the explanation for the dashed line to the figure legend and included the correlation scatter (standard deviation of the correlation results across the different target sites within the study regions); see the revised plot in Fig. 4 below. Yes, the dots in the plots are deliberately placed in the middle of the ring bin borders, i.e. at distances of 125, 375, 625, ... km, since each correlation value is an average value across a ring bin (from 0–250, 250–500, 500–750, ... km) by averaging the individual correlations obtained between the target site in the centre and all grid cells that lie within each bin.

Regarding the target sites your comment shows that there is a clear need for us to better explain this concept in the revised manuscript. We use "target site" as the term to denote a model grid cell from which we use the temperature time series ($T_{2m}$) to correlate all other grid cells and variables with, and which defines the centre of the ring bins. For a single target site we can study the average correlation with distance similar as shown in Fig. 5. But Fig. 5 involves a second averaging step: To improve statistics, and to obtain regional estimates, we use all model grid cells within our defined DML and Vostok regions as target sites, one after the other, to get the correlation–distance dependencies for each target site and for each variable ($T_{2m}$, $\delta^{18}O$, $\delta^{18}O^{(pw)}$). Then we average all these 26 (DML region) and 30 (Vostok region) curves that we obtain for each variable, respectively, to produce the curves shown in Fig. 5. The same approach is used for the results shown in Figs. 6 and A1.

We expanded the explanations in sections 2.3.1 and 2.3.3 (formerly 2.3.4) to clarify the concept of "target sites" and of "the averaging across target sites" to the reader. In addition, we revised the figure caption as follows:

"[…] Averaging was performed in two steps: first, for a given target site the correlations to the target site temperature were averaged across grid cells falling within 250 km wide consecutive rings around the given target site. Secondly, this analysis was conducted for all target sites in the DML (**a**) and Vostok (**b**) region and the results were averaged across the respective region (see Sects. 2.3.3 and 2.3.4 for details)."

[Figure]

**Figure 4.** Revised version of manuscript Fig. 5 with an estimated regional scatter of the correlation coefficient included.

Fig 6. These are interesting and show the clear difference for the precipitation weighted $\delta^{18}$O and T$_{2m}$ for DML, but why is the Vostok case relegated to the appendix? The fact that they are different is an interesting result.

We fully agree that the difference between the DML and Vostok regions is an interesting result. When writing up the manuscript, we were concerned that we could overload the results section with too many coloured contour plots, which is why we moved the Vostok plots to the appendix. However, we are happy to have the plots for both regions in the main text and we now combined the figures into a single Figure 6 and removed Appendix A.

Fig 7. The way this figure is arranged, is the top row, marked rank 5, that which has the max correlation, or is the rank 1 row the highest correlation row? I find it very curious that in the Vostok region the optimal arrangement comes with no local sites in many of the cases

The maximum correlation corresponds to the combination/row that is marked as rank 1; we added a clarifying remark to the figure caption.

Yes, it is indeed a curious result that in the Vostok region the optimal arrangement comes mostly without local sites, but previously we had seen no evidence for not trusting this result in view of the agreement between N = 3 and N = 5 and given the statistics from the large number of sampled locations in our ring sampling scheme. However, it now turned out that the outliers found in the climate model data (see below and the respective reply to reviewer #2) indeed caused some artefacts here. After removing those outliers, the updated Vostok results (below Fig. 5) appear much more consistent, including now the innermost (local) ring and being in general more consistent with the results of Fig. 6f (discussion paper figure A1c). We updated the paper figure with the new results and adjusted the results description:

- in the caption:

Displayed are subsets of the sampling correlation structures for $N\ell$ = 3 and 5, showing along the vertical axis the optimal five of all possible combinations of rings (best denoted as rank 1, fifth best as rank 5), i.e., those which exhibit the five highest mean correlation values across $10^5$ random trials of averaging $N\ell$ = 3 (**a**, **b**) or $N\ell$ = 5 (**c**, **d**) grid cells from these rings. […] Systematically, arrangements which combine ice cores from the innermost ring with ice cores further away are found to be optimal, with larger relative distances for the EDML target site.

- and in the main text:

Furthermore, We obtain similar results also when averaging $N\ell$ = 3 or 5 locations of the $\delta^{18}$O$^{(pw)}$ field

to reconstruct the target site temperature (Fig. 7). For computational reasons, we here only analyse single target sites. When EDML is set as the target site, the optimal sampling configuration is such that 1–2 locations lie in the innermost ring while the others are distributed at distances mostly between ~750 and 1500 km from the target. For reconstructing the Vostok target site temperature, the optimal locations combine the innermost ring with locations distributed mostly across the second to third (250–750 km) ring.

[Figure]

**Figure 5.** Update version of manuscript Fig. 7 showing the results after correcting the climate model data for artificial outliers.

line 220: Suggests that regional differences play a part here and I would like a comment on what those differences are (elevation, distance from coast etc?)

We can only speculate what possible reasons could there be for the difference in temperature–isotope correlation between the DML and Vostok regions (see Fig. 5). One factor might indeed be the larger elevation and distance from the coast, i.e. a stronger continentality at Vostok, and related to this, differences in the distillation paths of the transported vapour.

Line 232: Does the averaging have an effect on the significance of the correlations?

As pointed out above, there is statistically no sense in studying the significance of the correlation coefficients given that each time series has such a large number of data points.

Fig. 8 (b) I assume the colours on the histogram are the same as in (a), but please add the legend anyway. Can you comment on the low correlation outliers for the EDML case.

Yes, the colours are the same in (b) as in panel (a); nevertheless, we added a second legend to panel (b). The low correlation outliers in the EDML case stemmed from the grid cell combinations which included one anomalous grid cell located at ~ 72.4 °S, 22.5 °E; see Fig. 3. We had a deeper look into this and performed an outlier analyses on the climate model data, removing any artificial time series values from the grid cells. This also remedies the correlation outliers in the EDML histogram here. See our reply to a respective comment by reviewer #2 for more details on the outlier analysis.

Section 4.1 I found most of the discussion very clear and thorough, but re-iterate that a good schematic diagram to illustrate the length scales in the conceptual model would be very useful.

Thank you; please see our reply to your General Comments and Fig. 1 there.

References:

Casado, M., Münch, T., and Laepple, T.: Climatic information archived in ice cores: impact of intermittency and diffusion on the recorded isotopic signal in Antarctica, Clim. Past, 16, 1581–

1598, https://doi.org/10.5194/cp-16-1581-2020, 2020.

Laepple, T., Münch, T., Casado, M., Hoerhold, M., Landais, A., and Kipfstuhl, S.: On the similarity and apparent cycles of isotopic variations in East Antarctic snow pits, The Cryosphere, 12, 169–187, https://doi.org/10.5194/tc-12-169-2018, 2018.

Münch, T., Kipfstuhl, S., Freitag, J., Meyer, H., and Laepple, T.: Regional climate signal vs. local noise: a two-dimensional view of water isotopes in Antarctic firn at Kohnen Station, Dronning Maud Land, Clim. Past, 12, 1565-1581, https://doi.org/10.5194/cp-12-1565-2016, 2016.

Author Reply to the Review Comments by **Dmitry Divine (Referee #2)**

on the manuscript

**How precipitation intermittency sets an optimal sampling distance for temperature reconstructions from Antarctic ice cores**

by Thomas Münch, Martin Werner, and Thomas Laepple,
submitted to *Climate of the Past* (https://doi.org/10.5194/cp-2020-128).

Thank you very much, Dmitry Divine, for the time you spent on reading and reviewing our manuscript. Below we include a point-by point response to both the major and all minor comments. The original referee comments are set in normal black font, our replies in blue, and suggested changes to the manuscript are shown by citing the manuscript text on a gray background with changes in red.
* * *
**Major Comments:**

My major comment concerns the presentation of the sampling procedure in 2.3.2-2.3.3, which I have found not very straightforward to comprehend. One should admit I have spent quite some time trying to understand the actual details behind the technique, though this difficulty could of course be quite individual. The grip of understanding came later, while reading "Results", yet some questions still remain. A number of minor questions that emerged while reading the manuscript, could therefore be a result of my unclear understanding of the basics of the proposed method.

We are sorry and apologise for the fact that our description of the sampling procedure was not straightforward to comprehend and are grateful to the reviewer that he still spent the time trying to understand our approach. In addition to implementing the changes from our answers to the specific comments below, we thoroughly went through the overall methods text again and revised it completely in order to improve formulation and clarity. Furthermore, we included the following figure as an additional part of manuscript Fig. 1 to visualise our approach and hope that is helpful for aiding comprehension:

[Figure]

**Fig. 1** From an array of grid cells (grey points), we choose sets of grid cells from around a target site (black cross), which consist of $N_\ell$ cells and which are drawn from radial bins determined by selected combinations of rings (red). As an example for $N_\ell = 2$, possible grid cell sets are shown for the cases of (i) combining the innermost ring with itself (grid cells marked black), (ii) combining the innermost ring with the second ring (grid cells marked blue), and (iii) combining the third and the fourth ring (grid cells marked orange).

I would like to note also that sometimes the discussion around/use of terms like "target site" or "local site" may appear confusing, same as the actual dimension of the core network being discussed. May be some simplification/clarification of 2.3. can improve the readability?

The target site is always the site from which we use the pure temperature time series as a reference or, in other words, which is sought to be reconstructed from the "ice core" network. We clarified this in Sect. 2.3.1. "Local site" has been loosely referred to as a site close to the target site, so e.g. the target site grid cell itself or grid cells within the innermost ring. We revised the text either by directly specifying what we mean by "local site", or paraphrasing the term with other more appropriate wording. Finally, the "study regions" are areas from which we sequentially use all contained grid cells as target sites. The results with respect to each target site in a study region are then averaged across all target sites in order to arrive at regional estimates, such as for Figs. 5, 6, and A1. We clarified this in Section 2.3.3 (formerly 2.3.4).

In general, with respect to the sampling strategy, the question is why the authors initiated the procedure with these concentric rings used for spatial sampling, instead of just random seeding of the "sampling locations", calculating the metrics of interest and then ordering them according to the distances between the locations? It sounds way more straightforward to comprehend than via introducing these circular sampling areas with an increment of an arbitrary choice.

Yes, we agree that the approach of randomly seeding the sampling locations and subsequent ordering of the results according to the distances between the locations is, conceptually, a more straightforward procedure. In fact, our approach for N = 1, i.e. sampling one location only, is identical to the random seeding approach, if the latter approach uses a sufficient number of iterations to sample the entire required space.

However, this is also the critical point why we chose the different approach of the ring sampling scheme. While for N = 1 it is computationally easy and fast to sufficiently sample the required space (e.g. a 2000 km circle around a target site) by random seeding, this is more problematic for N >= 2. Here, the total number of possibilities of combining two, or more, grid cells is much larger than the actual number of grid cells (the more the larger N is), and a random seeding approach of these many possibilities will be strongly limited by available computation time. This will likely lead to an uneven sampling of the distance combinations, especially for distances farther away from the target site due to the radially increasing number of grid cells.

Our ring sampling approach circumvents this uneven sampling: i) Since we sample all possible ring bin combinations, we ensure to sample the entire available sampling space of (binned) distances relative to the target site. ii) For each ring combination, we either sample all possible grid cell combinations (N = 1 and N = 2), or we sample a fixed (but large) number of randomly chosen grid cell combinations (N > 2). This ensures that either the entire availble sampling space is actually sampled, or – at least – a fixed number of grid cell combinations for every ring bin combination, so that the expectation value for every ring bin combination builds on the same number of combined grid cells.

We added this motivation to the manuscript in section 2.3.2.

Also, as a suggestion for the future work, it would be highly useful to test the concept of this method on a different model with enabled stable water isotopes in precipitation in order to see how different the results/inference can be. Testing on the existing ice core network can be fairly problematic due to all the deficiencies (both in the available ice core and instrumental data) mentioned throughout the text.

This is a very good point. Indeed, we tested our concept also on the ECHAM6/MPI-OM-wiso pre-industrial control simulation (unpublished) and obtained comparable results. However, this might also be due to the fact that both models use the same isotope scheme. Therefore, the testing of our results with a completely different isotope-enabled climate model is needed. We added this suggestion to the conclusions of our manuscript.

**Minor Comments (the manuscript text shown in italics):**

Line 112: "*...define consecutive rings around this site with a 250 km radial width...*"

Here you refer to these concentric rings with a radius increment of 250 km, used for delimitation of the sampling regions, do I get it right? May be it needs to be specified already here. Can you also provide any rationale behind the value of 250 km?

Yes, we here refer to the concentric rings (red circles in Fig. 1) with 250 km radial increments which we use to sample grid cell combinations as a function of whether they lie within the same ring or within different rings. We refined the text here to make this clearer to the reader. We chose the value of 250 km radial extent as a trade-off between achieving a high spatial resolution and ensuring that a sufficient number of grid cells actually lie within the ring bin borders; e.g., the first ring (0–250 km) with respect to the EDML site includes already five grid cells only. Using a smaller radial extent (higher spatial resolution) would thus not be meaningful and would result in statistically less robust results. We added a clarifying remark to the caption of Fig. 1.

Line 118: "*Finally, we report the mean correlation for every ring combination by averaging across all correlations of the analysed grid-cell combinations.*" Is this averaging based on the distance between the locations, or just everything? How then the distance-based value is calculated?

The averaging is performed across all analysed grid cell combinations for a given ring combination. The distance information is then "only" given by the radial midpoint distances of the combined ring bins relative to the target site. We revised the entire section to improve the explanation of the ring sampling approach and also clarified how we obtain the distance information.

Line 119: "*...for sampling N locations from the model field depending on the distances between the locations.*" See my previous comment. If everything is averaged out, how the distance based sorting/ranking is implemented?

We clarified how we obtain the distance information; see our previous answer.

Line 136: *"(−78.47 S)*». No need in "-" before the latitude value if "S" is explicitly indicated.

Thanks for spotting this inconsistency; we corrected this.

Line 184: "*...depend on the specific simulated climate state or result...*"
It would be meaningful to add that it also includes the actual model used and the stable water isotope scheme applied in the model.

We added the possible dependence on the climate model and model isotope scheme here.

Line 212: "*...maximum average correlation is to sample one location from the innermost ring and the second location from the fifth ring*"
this is not entirely apparent as both maxima in Fig 6 seem to be found on the "5th ring".

We are afraid that this is a misunderstanding. There are indeed two maxima visible in Fig. 6c, both between 1000 and 1250 km (i.e. fifth ring), one along the x axis and one along the y axis. However, this is the same information since the locations of the two cores are indistinguishable, i.e., it doesn't matter whether we put the "first core" within the fifth ring and the "second core" within the first ring (maximum along the x axis) or vice versa. In other words, the figure is symmetric along the diagonal and half of the plot already contains the full information. We chose this way of presentation for aesthetic reasons. We added a sentence to the caption noting that the plot is mirrored along the diagonal for aesthetic reasons.

Line 255: "*For a conceptual model of the sampling correlation structure, we focus on three processes that influence...*". It is probably would be more relevant to write about focusing on three OF the processes that has an influence, as other processes are discarded in this conceptual model and this is mentioned in the text.

We edited the text as follows:

"For a conceptual model of the sampling correlation structure, we focus on the three main atmospheric processes that influence the oxygen isotope records in ice cores"

Line 280: "*When fixing one location to the target site and varying the distance from the target site of the second location...*"
This sentence appears again somewhat confusing to me. Do you actually average over "three" locations here or only two? You refer to fixing the core to the target site (first core), and then refer to the "second site". What then denotes "distance of first core " in the figures (like Fig 6)?

We are sorry for this ambiguity. We indeed average across only two locations. What is meant here in general is to fix one core (the "first core") to the innermost location and only vary the location of the second core. In the conceptual model, which is discussed at this point, the innermost location is identical to the centre of the rings, i.e. the target site (simply due to the fact how the conceptual model is set up numerically). In the climate model results (Figs. 6, A1), however, the innermost location in the ring sampling scheme can only be obtained for putting the first core within the first (innermost) ring (0–250 km). We revised the text accordingly here:

"When fixing the position of one core to the innermost location and varying only the distance from the target site of the second core location"

and at other respective passages to avoid this ambiguity.

Line 307: "*Our results which we obtained from analysing the climate model data and substantiated with our conceptual model provide guidance on where to drill N = 1,2,3 or more ice cores, or from which locations...*"
This statement is not entirely correct, the presented results tell about the relative distances (dimensions) of the core network optimal for the model, rather than point to specific locations that need to be derived via modelling for every target region.

We do not fully agree. If one believes in the "picking" results of directly analysing the best grid cell combinations (discussion paper Section 3.2 and Fig. 4), our results can directly advise where to drill cores for a specific target site. This is also elaborated in the following paragraph (discussion paper LL310–315). But you are correct that one would need to do the analysis for every specific target site one is interested in. In this regard, the ring sampling results are more general since they should apply to a larger region (DML, Vostok region) but with the downside that the results only tell us about the optimal relative distances of the core network, as you correctly observe. We revised the introductory paragraph of this section in order to better introduce these two different views, and noted the advantages and downsides of each approach in the following two paragraphs more clearly.

Line 311: "*However, it is unclear whether these results can be one-to-one transferred to the real world, since they might depend on dynamical processes in the atmosphere which could differ between climate states or depend on initial conditions.*" Consider adding "...or unaccounted model deficiencies"

We added this additional information.

Line 328: "*we expect the optimal spatial configuration to be more dependent on the study region*" ... and very likely on the GCMiso model used in the analysis.

Here, we talk specifically about the results for three or more ice cores (N >= 3); your statement rather applies to the results in general. We think it is thus sufficient to revise the text around L184, as suggested above in the respective comment.

Line 331:" *We thus need to create an isotope record that*" Consider adding "As a proof of concept"

We added the suggested phrase.

Line 352: *...we expect similar results to hold for other parts of Antarctica, and potentially also for other large-scale ice-coring regions such as Greenland*"
One can add that this is conditional on a simplified assumption of a nearly anisotropic exponential decorrelation scale length to be valid

We added this limitation.

Figure 3: Why the correlation value for a cell at approximately 70 S and 20E stands out?

Inspection of the time series of this grid cell located at ~ 72.4 °S, 22.5 °E shows that the isotope time series there exhibits one anomalous time step at the model year 970 CE where the delta value erroneously rises far above 0 permil. This causes the observed outlier correlation with the temperature time series. Based on this finding, we made a systematic investigation of outlier values in the isotope time series across all model grid cells, defining outliers as annual values which lie above or below a threshold of 4 times the time series' interquartile range. With this approach, we found in total 443 annual outlier values (236 for $\delta^{18}O$ and 207 for $\delta^{18}O^{(pw)}$), which nearly all occur for the model year 970 CE and with the above mentioned outlier at ~ 72.4 °S, 22.5 °E being by far the most prominent one. Such spikes in the modelled isotopic data are due to numerical instabilities which can occur in very dry regions.

We removed all these anomalous time steps from the model data and re-ran all paper analyses for the revised version, updating all plots and all text passages which quote specific results. We find that the outliers haven't had any effect on the previous general results of our paper and their removal thus does not change our general conclusions, but we see some overall improvement in temperature–isotope correlation after outlier removal, and the most notable difference is that the optimal ring combinations for Vostok for N = 3 and N = 5 now always include the innermost ring (Fig. 7b, d), which makes these results actually more consistent with the results from Fig. 6f.

Figure 6: The caption is somewhat confusing. Is the "target site" also to be sampled or not? If this is the case, should this be a 3-dimensional case or not?

No, it is the 2-dimensional case, i.e. averaging two locations, as explicitly stated in the first and second sentence of the caption. The grid cell of the target site lies in the centre of the innermost ring, so it is (implicitly) included in the analyses when sampling is performed for combining the innermost ring with itself (distance of both cores <250 km) or with one of the other rings (distance of only one of the cores <250 km). But please note that the results are the average across all grid cell combinations for a specific ring combination, hence the target site grid cell only contributes proportionately to the overall average value for the combinations which include the innermost ring.

We improved the description in the caption as follows:

"[…] Shown is the mean correlation of all possible single correlations to the target site temperature of the average of two grid cells of (**a**, **d**) $T_{2m}$, (**b**, **e**) $T_{2m}^{(pw)}$ and (**c**, **f**) $\delta^{18}O^{(pw)}$ time series sampled from the same ring or from two different rings. This analysis was conducted for every target site in the DML region (panels **a**–**c**) and in the Vostok region (panels **d**–**f**) and the results were then averaged across the respective region."

Figure 7: what is "Rank" on y-axis? Ranking according to the maximum correlation attained? It should than be mentioned explicitly.

Yes, the "Rank" means ranking according to the maximum attained correlation, with Rank 1 denoting the case with the highest correlation. We added this information to the figure caption.

Figure B1, caption.
*"Note that the plots (a) and (c) are based on the same parameters and therefore identical".*
Why and where they are identical? This is not evident from the plots.

Thanks for spotting this typo, correct is that plots (a) and (d) are identical. We corrected this.

on the manuscript

**How precipitation intermittency sets an optimal sampling distance for temperature reconstructions from Antarctic ice cores**

by Thomas Münch, Martin Werner, and Thomas Laepple,
submitted to *Climate of the Past* (https://doi.org/10.5194/cp-2020-128).

Dear Zoltán Kern and István Hatvani,

thank you very much for posting a short comment on our manuscript, to which we include a reply below. For this, we repeat your original comment, set in normal black font, and add our replies in blue.
* * *
Dear Authors,

First of all let us congratulate you on this very concise and precisely documented study which was highly interesting to read.

Thank you very much.

In the following we would like to comment on two key results.

We agree that ambient temperature of precipitation events should be expected to show a stronger correlation to precipitation d18O than annual mean temperature due to the intermittency of precipitation. We also agree that the data calculated from the simulation results reflect this theoretical relationship, however we would like to note that in a more experimental approach (http://journals.pan.pl/dlibra/publication/116059/edition/100870/content) we tested this idea and an opposite result was obtained. We found that amount weighting is incapable of ameliorating the signal replication between the stations and the ice cores, while arithmetic means gave the stronger linear relationships. The explanation is thought to be isotopic exchange between vapor and surface snow. In the present paper this may open an additional perspective from which the contrast seen in Figure 3 and Sects. 3.1 and 4 can be viewed from.

Thank you for pointing us to this study, which constitutes an important test of the effect of precipitation intermittency on the correlation between ice-core derived isotopic composition and temperature based on real world data.

However, we do not fully agree with your interpretation. In fact, when only data from weather stations is concerned, the study shows the expected result that, firstly, the precipitation-weighted temperature shows a higher degree of correlation with the temporal variations of the precipitation isotopic composition (which you do acknowledge in the study), and, secondly, that this effect is more pronounced at a more continental site, where precipitation is more intermittent, as compared to a maritime site on the Antarctic Peninsula with a rather regular seasonal precipitation distribution. These results are thus in line with our results from the climate model that in general the precipitation-weighted temperature correlates to a higher degree with the isotopic composition (see our Fig. 3), but not so much for coastal sites (see the respective map for the difference in correlation in our reply to reviewer #1).

The problem arises when the real ice core data is concerned, which in your study may actually exhibit a lower correlation with the weighted than with the unweighted station temperature records. You interpret this finding in terms of a possible surface–atmosphere exchange of vapour, which might lead to a more regular isotopic temperature signal imprinted into the ice core record. However, we think that also other factors might explain the observed lower correlation. Firstly, the significance of the correlation values is unclear, especially given the small temporal overlap of the data (e.g. Fig. 3 in the study). Secondly, there is quite some distance between the sites of the used ice cores and weather

stations, and it remains unclear whether these sites really exhibit similar vapour sources and trajectories, etc. Thirdly, and probably most important in our opinion, you do not take into account stratigraphic noise, which can strongly influence the isotopic variability and thus the correlation values, especially for the lower accumulation sites. A comparison with the correlation values between the ice cores could help here to estimate the signal-to-noise ratio.

Overall, we do agree that surface–atmosphere vapour exchange might partially counteract the impact of precipitation intermittency on the recorded temperature signal, if it constitutes a significant contribution to the snow isotopic composition[*]. However, as long as the *atmospheric* isotope signal is concerned, this does not affect the notion that ice core sites should be combined in a way as shown in our manuscript in order to optimally avoid the impact of precipitation intermittency.

[*] The same applies actually to diamond dust (clear sky) precipitation, which is also more regular than convective-type precipitation and which might be under-represented in General Circulation Models.

We also agree with the concept that the signal can be enhanced by averaging isotope records across space, however it is quite strange that ". . .the optimal sampling strategy is to combine a local ice core with a more distant core 500–1000km away. A similarly large distance between cores is also optimal for reconstructions that average more than two isotope records." In this paper http://dx.doi.org/10.1016/j.polar.2017.04.001 we performed geostatistical analysis of 60 ice core derived d18O time series in Antarctica to determine their spatial autocorrelation structure and to find the area yet unrepresented by the assessed set of records. The spatial autocorrelation (varography; Matheron 1963) is not equivalent to decorrelation (Appendix B1-2) but also measures the spatial similarity of the studied parameter. For instance, we obtained a 350km spatial "influence" range of the assessed ice core d18O records via semivariogram analysis, which would be interesting to be compared with your results regarding the question: Why are the original ice core d18O data spatially correlated within 250km and the modeled ones in your study above 500 km to simplify the question...

The variogram technique is indeed an interesting alternative means to study the spatial similarity in the data – thank you for pointing us to this. But we think again that the respective results in your study do not conflict with our findings. To conduct the variogram analysis on the ice core data, you first remove by a multivariate regression the influence of elevation, coastal distance and longitude. Since the first two terms are also the main driving variables on the temperature field, the resulting $\delta^{18}O$ residuals exhibit variations which should not be related to the large-scale temperature variations. Thus, the spatial range of influence of 350 km, which you find in the variogram, could be the result of isotopic variability driven by spatially coherent precipitation variations, in line with the same-order-of-magnitude precipitation decorrelation scales that we find in model data (see our Appendix B5) and which we use as decorrelation scale (500 km) to model the intermittency noise. As you mention in your study, an alternative interpretation for the 350 km scale could be regional-scale temperature variations imprinted into the isotopic composition of surface snow by vapour exchange processes. However, in both the climate model and our conceptual model, vapour exchange is not taken into account, so that the regional-scale coherence of the residual isotopic variations is only driven by precipitation variability. Then, it makes perfect sense to place ice cores farther apart than this scale (but below the decorrelation scale of the temperature field), since this optimally averages out the noise by precipitation variability and thereby maximises the correlation with temperature.

We note, however, that in the real world the observed spatial range of influence in the ice core data could be a combined result of both coherent precipitation variability and vapour exchange. Then, the relevance of our results for actual ice core studies would depend on the relative contributions of both processes. To make progress here, we ultimately need solid quantitative estimates of the importance of vapour exchange processes across temporal scales, at least for the main ice coring regions.

These experimental findings based on real life data might worth consideration when your model results are evaluated and may also serve as a good addition to the discussion.

Overall, we agree that the effect of potential vapour exchange between the atmosphere and the surface snow might influence the relevance of our results for real world applications, but this heavily depends on the actual strength of the effect. We added a paragraph about these issues to the end of the discussion section of the manuscript.